# Mechanisms underlying the response of mouse cortical networks to optogenetic manipulation

**Alexandre Mahrach[1], Guang Chen[2], Nuo Li[2], Carl van Vreeswijk[1], David Hansel[1]\***

[1]CNRS-UMR 8002, Integrative Neuroscience and Cognition Center, Paris, France; [2]Department of Neuroscience, Baylor College of Medicine, Houston, United States

**Abstract** GABAergic interneurons can be subdivided into three subclasses: parvalbumin positive (PV), somatostatin positive (SOM) and serotonin positive neurons. With principal cells (PCs) they form complex networks. We examine PCs and PV responses in mouse anterior lateral motor cortex (ALM) and barrel cortex (S1) upon PV photostimulation in vivo. In ALM layer five and S1, the PV response is paradoxical: photoexcitation reduces their activity. This is not the case in ALM layer 2/3. We combine analytical calculations and numerical simulations to investigate how these results constrain the architecture. Two-population models cannot explain the results. Four-population networks with V1-like architecture account for the data in ALM layer 2/3 and layer 5. Our data in S1 can be explained if SOM neurons receive inputs only from PCs and PV neurons. In both four-population models, the paradoxical effect implies not too strong recurrent excitation. It is not evidence for stabilization by inhibition.

**\*For correspondence:**
dhansel0@gmail.com

**Competing interests:** The authors declare that no competing interests exist.

## Introduction

Local cortical circuits comprise several subclasses of GABAergic interneurons which together with the excitatory neurons form complex recurrent networks (*Goldberg et al., 2004*; *Jiang et al., 2015*; *Karnani et al., 2016*; *Markram et al., 2004*; *Moore et al., 2010*; *Pfeffer et al., 2013*; *Tasic et al., 2018*; *Tremblay et al., 2016*). The architecture of these networks depends on the cortical area and layer (*Beierlein et al., 2003*; *Jiang et al., 2013*; *Rudy et al., 2011*; *Xu et al., 2013*; *Xu and Callaway, 2009*).

Optogenetics is now classically used to reversibly inactivate a particular cortical area or neuronal population to get insights into their functions (*Atallah et al., 2012*; *Guo et al., 2014b*; *Lee et al., 2012*; *Li et al., 2015*; *Svoboda and Li, 2018*). Optogenetics has also been applied to isolate the different components (e.g. feedforward *vs.* recurrent) of the net input into cortical neurons (*Lien and Scanziani, 2018*; *Lien and Scanziani, 2013*). It can also be used to experimentally probe the architecture of local cortical circuits (*Moore et al., 2018*; *Xu et al., 2013*). However, because of the complexity of these networks and of their nonlinear dynamics, qualitative intuition and simple reasoning (e.g. 'box-and-arrow' diagrams) are of limited use to interpret the results of these manipulations.

'Paradoxical effect' designates the phenomenon that stimulation of a GABAergic interneuron population not only decreases the average activity of the principal cells (PCs) but also decreases the activity of the stimulated population (*Murphy and Miller, 2009*; *Ozeki et al., 2009*; *Tsodyks et al., 1997*). Intuitively, paradoxical effect arises when the stimulation induces a strong activity suppression in the PCs (*Kato et al., 2017*; *Moore et al., 2018*), such that the overall (synaptic+stimulus) excitation to the stimulated population decreases. However, the precise conditions under which the paradoxical effect occurs are difficult to establish without mathematical modeling.

In simple models consisting of only two populations (one excitatory and one inhibitory) these conditions have been mathematically derived. The paradoxical effect occurs when the networks

operates in the regime known as *inhibition stabilized* (inhibition stabilized networks, ISN) in which the total the total recurrent excitation is so strong that inhibition is necessary to prevent a blow up in the activity (*Murphy and Miller, 2009*; *Ozeki et al., 2009*; *Tsodyks et al., 1997*). Networks, with several inhibitory populations have been recently investigated (*Garcia Del Molino et al., 2017*; *Litwin-Kumar et al., 2016*; *Sadeh et al., 2017*). These studies considered network models with synaptic currents small compared to neuronal rheobase currents (*Gerstner et al., 2014*; *Lapicque, 1909*). However, interactions in cortex are stronger than what is assumed in these studies (*Shadlen and Newsome, 1994*).

Simple networks with strong interactions comprising one excitatory and one inhibitory population have been studied extensively. In a broad parameter range not requiring fine-tuning, such networks dynamically evolve into a state in which strong excitation is balanced by strong inhibition such that the *net* input into the neurons is comparable to their rheobases (*van Vreeswijk and Sompolinsky, 1998*; *van Vreeswijk and Sompolinsky, 1996*). The theory of balanced networks has been developed for a variety of single neuronal models including binary neurons (*van Vreeswijk and Sompolinsky, 1998*; *van Vreeswijk and Sompolinsky, 1996*), rate models (*Harish and Hansel, 2015*; *Kadmon and Sompolinsky, 2015*), leaky-integrate-and fire neurons (*Hansel and Mato, 2013*; *Mongillo et al., 2012*; *Rosenbaum and Doiron, 2014*; *Roxin et al., 2011*; *Van Vreeswijk and Sompolinsky, 2005*) and conductance-based models (*Hansel and van Vreeswijk, 2012*; *Pattadkal et al., 2018*).

In the present study, we investigate experimentally the effects of the photostimulation of PV interneurons on the anterior lateral motor cortex (ALM) and barrel cortex (S1) of the mouse. We show that two-population network models do not suffice to account for these effects. To overcome this limitation, we develop a theory for the paradoxical effect in balanced networks that takes into account the multiplicity of GABAergic neuronal populations. Combining analytical calculations and numerical simulations, we study the responses of these networks at population and single neuron level. For two-population balanced networks it has been shown that the paradoxical effect only occurs when the network is inhibition stabilized (*Pehlevan and Sompolinsky, 2014*; *Wolf et al., 2014*). Here we show that in contrast, in four-population networks, the paradoxical effect can occur even if the network is not inhibition stabilized. We conclude with prescriptions for experiments that according to the theory can be informative about network architectures in cortex.

## Results

### ALM layer 5 and S1 exhibit paradoxical effect but not ALM layer 2/3

We expressed a red-shifted channelrhodopsin (ReaChR) in PV interneurons to optogenetically drive local inhibition in the barrel cortex (S1) and anterior lateral motor cortex (ALM) of awake mice (*Hooks et al., 2015*). We used orange light (594 nm) to illuminate a large area of ALM or S1 (2 mm diameter), photostimulating a large proportion of PV interneurons (*Figure 1A*). We measured the light-induced effects on neural activity using silicon probe recordings. In both brain areas, putative PCs and putative PV neurons were identified based on spike width (Methods). Neurons with wide spikes were likely mostly PCs. Units with narrow spikes were fast spiking (FS) neurons and likely expressed parvalbumin (*Cardin et al., 2009*; *Guo et al., 2014b*; *Olsen et al., 2012*; *Resulaj et al., 2018*). We investigated the responses of these neurons as a function of the photostimulation intensity in ALM layer 2/3 and layer 5, and in S1.

We found that in all recorded layers and areas, the population average activity of the PCs decreased with the optogenetic drive (*Figure 1B*, *Figure 2*). In contrast in ALM, the PV population exhibited a behavior which depended on the recorded layer.

In ALM layer 2/3, the population average firing rate of PV neurons monotonically increased with the photostimulation intensity. However, individual neuron responses were heterogeneous. Most PV neurons increased their spike rates from baseline with increased photostimulation intensity. Some PV neurons initially decreased their spike rates below baseline for low light intensity.

In ALM layer 5, the response of the PV population was non-monotonic. For low laser intensity, their activity paradoxically decreased with the optogenetic drive. The slope of the normalized firing rate *v.s.* laser intensity was significantly different from zero for both the PC and PV populations (*Figure 1F*). The ratio of their slopes was $0.62 \pm 0.28$. At high photostimulation intensity, the activity

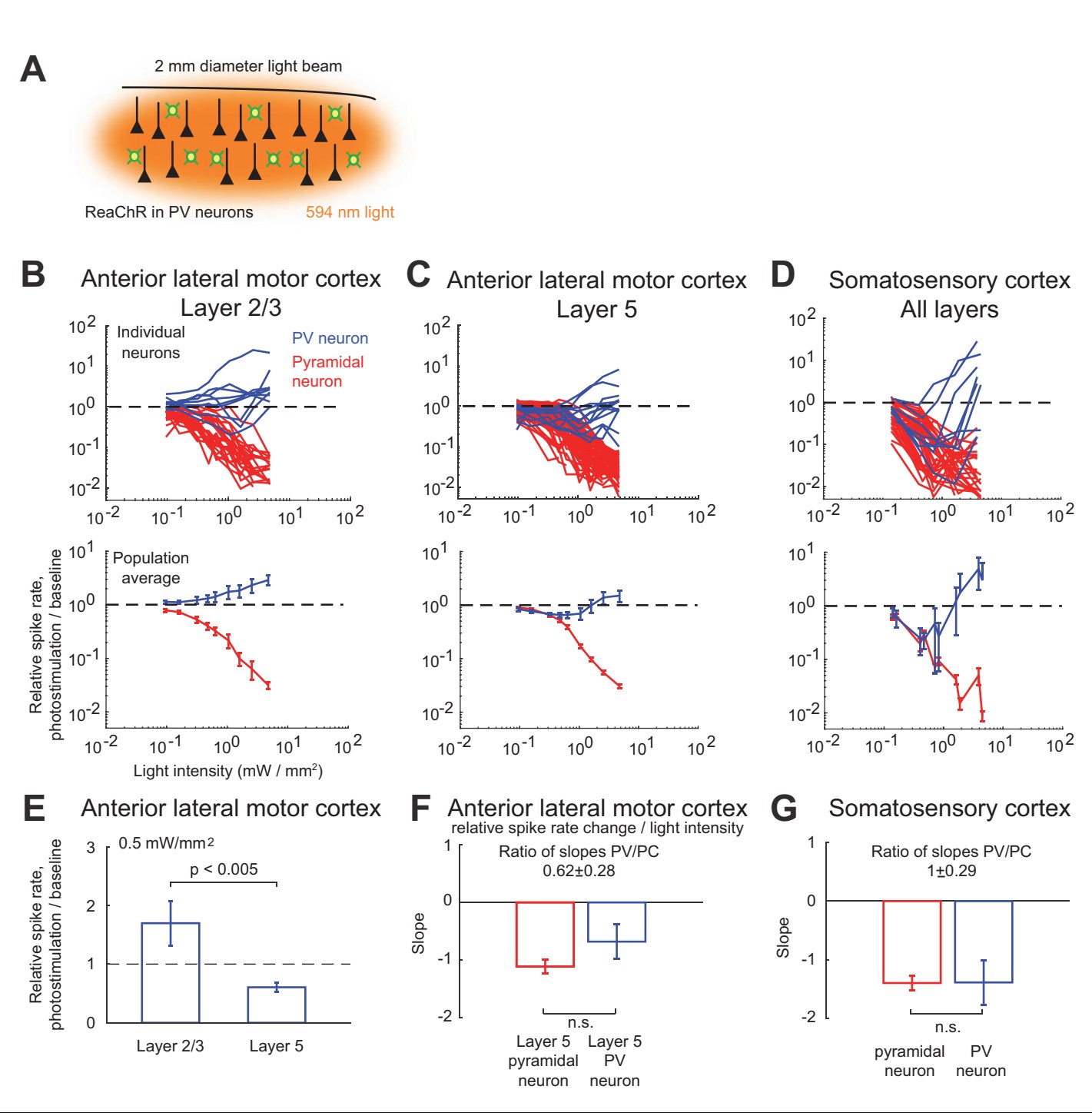

**Figure 1.** Effects of photostimulation of PV-positive interneurons in the mouse neocortex. (**A**) Scheme of the experiment. (**B–C**) Normalized spike rate as a function of laser intensity in different layers and brain areas. Top, individual neuron responses of the PCs (red) and PV (blue) neurons; bottom, population average responses. (**B**) ALM: layer 2/3: n = 26 (PCs), n = 9(PV); (**C**) ALM layer 5: n = 62 (PCs), n = 12 (PV). (**D**) S1: n = 52 (PCs), n = 8 (PV). Mean ± s.e.m. across neurons, bootstrap. (**E**) Comparison of PV neurons' normalized spike rates between ALM Layer 2/3 and Layer five at laser intensity 0.5 mW/mm$^2$. (**F**).Slope of PCs and PVs' normalized spike rate as a function of laser intensity. Data from ALM layer 5. Slopes are computed using data from 0.3 mW/mm$^2$ and below, before the spike rate of PV neurons begin to increase. Mean ± SEM, bootstrap (Methods). (**G**) Same as (**F**) but for data from S1. In (**F** and **G**) the difference between the slopes for the PC and PV populations is not significant.

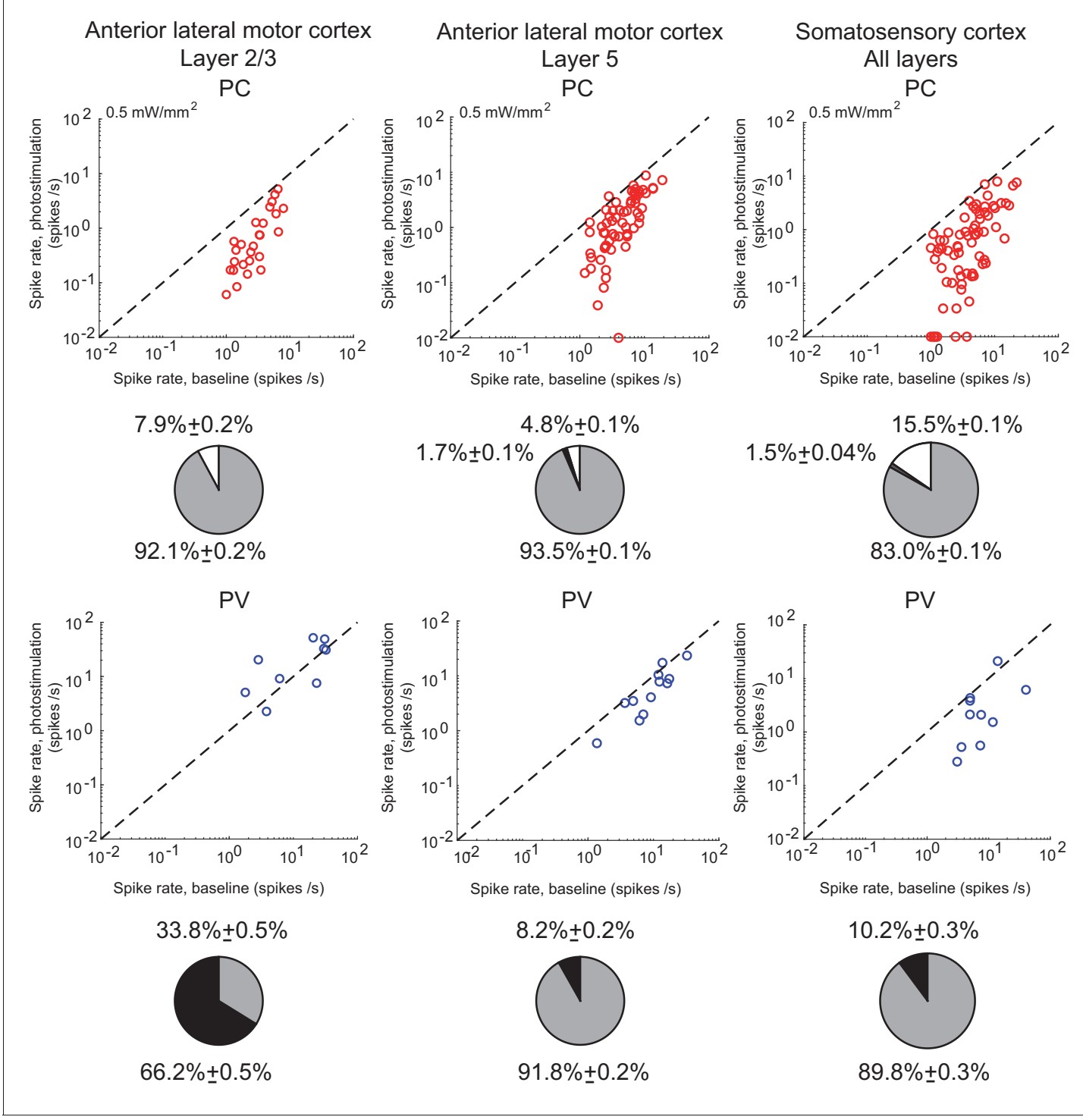

**Figure 2.** Spike rates of PCs (top) and PV neurons (bottom). Dots correspond to individual neurons. Laser intensity is 0.5 mW/mm². Pie charts represent the fraction of neurons with different types of changes. Mean ± s.e.m. bootstrap. Black, fraction of neurons with activity increase larger than 0.1 Hz. Light gray, fraction of neurons with activity decrease larger than 0.1 Hz. Dark gray, fraction of neurons with activity change smaller than 0.1 Hz. White, fraction of neurons with activity smaller than 0.1 Hz upon PV photostimulation.

of the PV population increased. At intermediate photostimulation intensity (0.5 mW/mm$^2$), the response of the PV neurons was significantly different between layer 2/3 and layer 5 (*Figure 1E*, p<0.005, unpaired t-test, two-tailed test).

Paradoxical decrease in PV neurons activity with the optogenetic drive was also observed in S1. Remarkably, the concomitant decrease of the PC and the PV population activities was proportional (*Figure 1G*, ratio of slopes PV/PC, mean ± SEM; S1, 1 ± 0.29).

In both ALM layer 5 and S1, there was also a large diversity of responses. Most PV neurons decreased their activity at low photostimulation intensity. At high laser intensity (5 mW/mm$^2$), a fraction of PV neurons (6/12 in ALM layer 5 and 6/10 in S1) had a larger response than baseline, while the rest remained suppressed. *Figure 2* shows the spike rates of PCs and PV neurons at an intermediate light intensity (0.5 mW.mm$^{-2}$).

## Network models

To assess the network mechanisms which may account for the experimental data from ALM and S1, we first considered models consisting of one excitatory and one inhibitory population. Since it is well established that cortical circuits involve a variety of inhibitory subpopulations, we later extended the theory to network models of four populations of neurons representing PCs and three subtypes of GABAergic interneurons in cortex. In all our models, neurons are described as integrate-and-fire elements. The data we seek to account for, were obtained in optogenetic experiments in which the laser diameter was substantially larger than the spatial range of neuronal interactions and comparable to the size of the area in which activity was recorded. Therefore, in all our models, we assume for simplicity that the connectivity is unstructured. We modeled the ReachR-optogenetic stimulation of the PV population as an additional external input, $I_{opto}$, into PV neurons. We assumed that it depends on the intensity of the laser, $\Gamma_{opto}$, as $I_{opto} = I_0 log\left(1 + \frac{\Gamma_{opto}}{\Gamma_0}\right)$ where $I_0$ and $\Gamma_0$ are parameters (*Figure 3— figure supplement 1*; *Hooks et al., 2015*).

## Two-population model

The two-population network is depicted in *Figure 3A*. It is characterized by four recurrent interaction parameters, $J_{\alpha\beta}$, and two feedforward interaction parameters, $J_{\alpha 0}$, $\alpha, \beta \in \{E, I\}$ (see Materials and methods).

Results from numerical simulations of the model are depicted in *Figure 3B* and C where, the dependence of the population activities normalized to baseline, are plotted against the intensity of the laser, $\Gamma_{opto}$. *Figure 3B* shows the response of the network where the recurrent excitation, $J_{EE}$, is non zero. The activity of the PV population, $r_1$ varies non-monotonically with the laser intensity. For small intensities, $r_1$ paradoxically decreases together with the activity of the PCs, $r_E$. This paradoxical effect stems from the fact that the decrease in the activity of the PCs yields a reduction in the excitation to PV neurons which is not compensated for by the optogenetic drive. As a result, the net excitation to PV neurons diminishes yielding a decrease in $r_I$. When $r_E$ becomes very small, this mechanism does not operate anymore and consequently, $r_I$ increases as $\Gamma_{opto}$ is increased further. In *Figure 3C*, $J_{EE}$ is zero, $r_I$ monotonically increases with the light intensity whereas $r_E$ monotonically decreases. For small intensities, $r_I$ is close to a constant. It starts to increase appreciably only when $r_E \simeq 0$. Therefore, the PV response is not paradoxical.

Qualitatively this model seems to account for our experimental data from ALM layer 2/3, ALM layer 5 and S1. It would imply that in layer 5, $J_{EE}$ is sufficiently large to generate the paradoxical effect, while in layer 2/3 this is not the case. On closer inspection however, there are major discrepancies between the simulation results and the experimental data. In our recordings in both ALM layer 5 and S1, the PV population activity reaches a minimum while the PCs are still significantly active: relative to baseline the activity is 40% in ALM and 25% in S1. In contrast, in the two-population model, the minimum of the PV activity is reached (Appendix 1B) when excitatory neurons are virtually completely silenced (*Figure 3B*, *Figure 3—figure supplement 2A*). In fact one can show that for sufficiently large K, when $r_I$ is minimum, the activity of the excitatory population is exponentially small in K. As a result, to account for the data one needs to assume that $K \simeq 10$.

In addition, in the experimental data the activities of the PC and PV populations in S1 decrease in equal proportions before the minimum of the PV activity (*Figure 1B*). This cannot be accounted for in a two-population model unless parameters are fine-tuned (*Figure 3—figure supplement 3*).

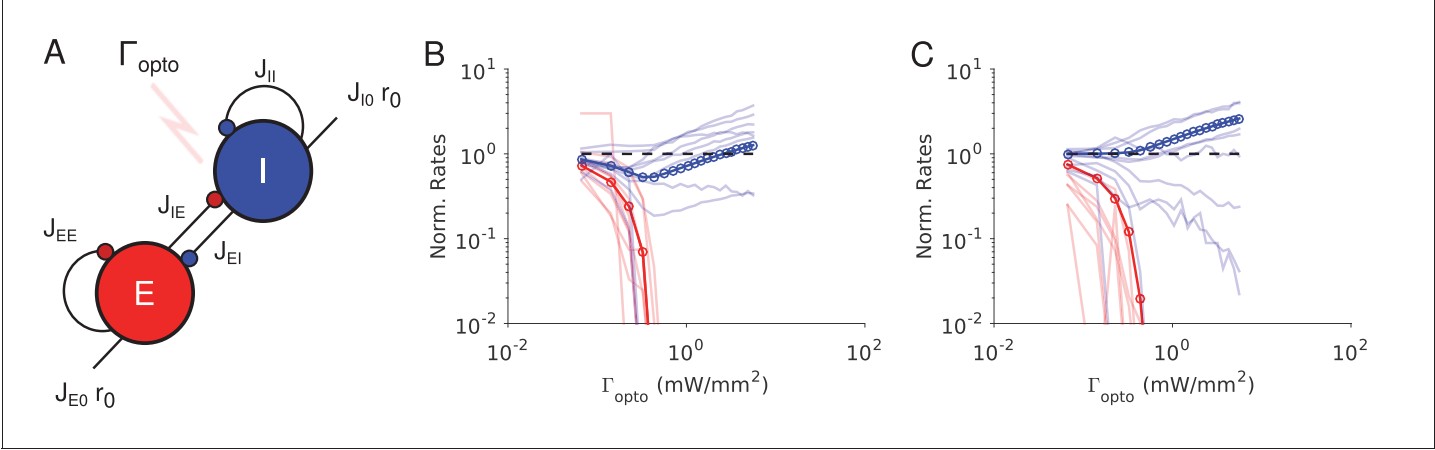

**Figure 3.** Paradoxical effects in the two-population model. (**A**) The network. (**B–C**) Responses of PCs and PV neurons normalized to baseline *vs.* the laser intensity, $\Gamma_{opto}$, for different values of the recurrent excitation, $j_{EE}$. (**B**) $j_{EE} = J_{EE}/\sqrt{K}$, the network exhibits the paradoxical effect. (**C**) $j_{EE} = 0$, the population activity of PV neurons is almost insensitive to small laser intensities. Red: PCs. Blue: PV neurons. Thick lines: population averaged responses. Thin lines: responses of 10 neurons randomly chosen in each population. Firing rates were estimated over 100s. Parameters: $N_E = 57600$, $N_1 = 19200$, $K = 500$ $N_1 = 19200$. Other parameters as in *Tables 1–2*. Baseline firing rates are: $r_E = 5.7Hz$, $r_I = 11.7Hz$ (**B**) and $r_E = 1.5Hz$, $r_I = 5.7Hz$ (**C**). At the minimum of $r_I$ in (**B**), $r_E = 0.06Hz$.

The online version of this article includes the following figure supplement(s) for figure 3:

**Figure supplement 1.** Current, $I_{opto}$, *v.s.* laser intensity, $\Gamma_{opto}$.

**Figure supplement 2.** Effects of $K$ on the responses of a two-population network to photoactivation of the inhibitory population.

**Figure supplement 3.** Two-population model.

Analytical calculations (Appendix 1B) supplemented with numerical simulations show that this proportional decrease only happens when the determinant of the interaction matrix, $J_{\alpha\beta}$, is close to zero. Moreover, the external input must also be fine-tuned so that the neurons have biologically realistic firing rates (*Figure 3—figure supplement 3*).

The experimental data from ALM layer 2/3 show that for already small light intensity the activity of PV neurons increases appreciably. This is in contrast with *Figure 3C*. In *Figure 3—figure supplement 2B*, we show that the two-population model can account for this feature only if the recurrent excitation is very weak in that layer and the connectivity is extremely sparse.

These discrepancies prompted us to investigate whether models with several populations of inhibitory neurons can account for our experimental data without fine-tuning. We focus on two four-population network models. Both consist of three populations representing PCs, PV and SOM neurons and a fourth population representing other inhibitory neurons. The main difference between the two models lies in the inhibitory populations from which SOM neurons receive inputs.

## A four-population model with V1-like architecture (Model 1)

We first investigated the dynamics of a four-population network with an architecture that is similar to the one reported in layer 2/3 in V1 (*Pfeffer et al., 2013*) and S1 (*Lee et al., 2013*) (*Figure 4A*). The model consists of four populations representing PCs, PV, SOM and VIP neurons. SOM neurons do not interact with each other (*Adesnik et al., 2012*; *Gibson et al., 1999*; *Hu et al., 2011*). VIP neurons only project to the SOM population (*Jiang et al., 2015*; *Pfeffer et al., 2013*). All neurons

**Table 1.** Connection strength matrix (rows: postsynaptic populations; columns: presynaptic populations).

| $J_{\alpha\beta}(\mu A.ms.cm^{-2})$ | Feedforward | PC | PV |
|---|---|---|---|
| PC | 17 | 29 | 30 |
| PV | 17 | 36 | 36 |

Parameters of the two-population model.

**Table 2.** Synaptic time constants.

| $\tau_{\alpha\beta}(ms)$ | E | I |
|---|---|---|
| E | 4 | 2 |
| I | 2 | 2 |

Default parameters of Model 1.

except SOM receive inputs from sources external to the network (*e.g.* thalamus) (*Beierlein et al., 2003*; *Beierlein et al., 2000*; *Cruikshank et al., 2010*; *Ma et al., 2006*; *Xu et al., 2013*). The same architecture was considered in *Litwin-Kumar et al. (2016)*.

Following *Pfeffer et al. (2013)*, the PV population does not project to the SOM population. Other studies have reported such a connection (*Jiang et al., 2015*). However, adding such a connection to Model 1 does not qualitatively affect the PC and PV responses (see Appendix 1C).

We considered parameter sets such that: 1) At baseline, the network is operating in the balanced state with all populations active; 2) the activity of the PC population decreases with the laser intensity as observed in our experiments.

## Theory in the large *N, K* limit

It is instructive to consider the limit in which the number of neurons in the network, *N*, and the average number of connections per neuron, *K*, go to infinity. In this limit, the analysis of the stationary

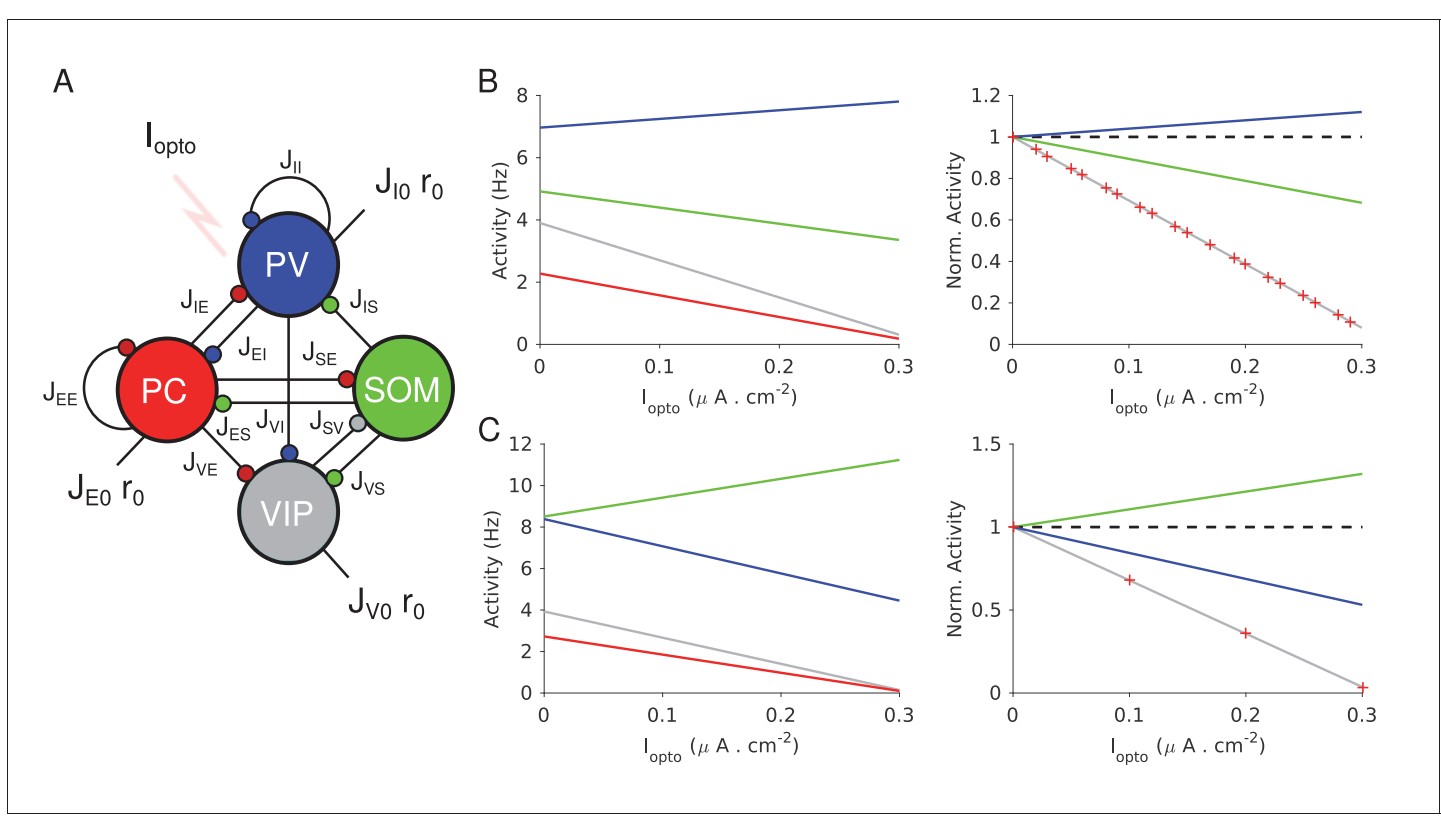

**Figure 4.** Population activities *vs.* $I_{opto}$ in Model 1 in the large *N, K* limit. (**A**) The network is composed of four populations representing PCs, PV, SOM and VIP neurons. The connectivity is as in *Pfeffer et al. (2013)*. (**B**) Parameters as in *Table 4*. The activity of PV cells increases with $I_{opto}$ while for the three other populations it decreases. (**C**) Parameters as in *Table 5*. The activity of SOM neurons increases with $I_{opto}$ while for the three other populations it decreases. Right panels in B and C: the activities are normalized to baseline.

The online version of this article includes the following figure supplement(s) for figure 4:

**Figure supplement 1.** Graphical representation of the population susceptibilities upon stimulation of PV in Model 1 (large *N, E* limit).

**Figure supplement 2.** Population activities *vs.* $I_{opto}$ in Model 1 (large *N, K* limit).

state of the network simplifies (see Materials and methods). This stems from the fact that when interactions are numerous, excitatory and inhibitory inputs are strong and only populations for which excitation is balanced by inhibition have a finite and non-zero activity. The average activities of the four populations are then completely determined by four linear equations, *the balance equations*, which reflect this balance. Solving this system of equations yields the population activities, $r_\alpha$, $\alpha = E$, $I$, $S$, $V$, as a function of the external inputs to the network. In particular, when the laser intensity is sufficiently small, the four populations are active and their firing rates vary linearly with the current induced by the photostimulation (Appendix 1C).

*Figure 4* plots the activities of the populations *vs.* the optogenetic input into PV neurons, $I_{opto}$, for two sets of interaction parameters. In *Figure 4B*, the activity of the PV population, $r_I$, increases with $I_{opto}$. In contrast, in *Figure 4C*, $r_I$ decreases with $I_{opto}$: the response of the PV population is paradoxical.

To characterize for which interaction parameters the PV response is paradoxical, we consider the $4 \times 4$ susceptibility matrix $[\chi_{\alpha\beta}]$. The element $\chi_{\alpha\beta}(\alpha, \beta = E, I, S, V)$ is the derivative of the population activity, $r_\alpha$, with respect to a small additional input, into population $\beta$, $I_\beta$. Evaluated for small $I_\beta$, $\chi_{\alpha\beta}$ characterizes by how much $r_\alpha$ varies with an increasing but weak extra input into population β. Its sign indicates whether $r_\alpha$ increases or decreases with $I_\beta$. The elements of the susceptibility matrix can be decomposed in several terms corresponding to the contributions of different recurrent loops embedded in the network (Appendix 1C). Using this decomposition one can show whether the PV response is paradoxical or not depends on the interplay between two terms. One is the gain of the disinhibitory feedback loop PC-VIP-SOM-PC and the other is the product of the recurrent excitation, $J_{EE}$, with the gain of the disinhibitory feedback loop VIP-SOM-VIP (*Figure 4—figure supplement 1*). Remarkably, PV neurons are not involved in these two terms. A straightforward calculation (*Equation A37*) then shows that the response of PV neurons increases with $I_{opto}$ if the recurrent excitation is sufficiently strong, namely if

$$J_{EE} > J_{EE}^* = J_{VE}J_{ES}/J_{VS} \tag{1}$$

The denominator in $J_{EE}^*$ is the strength of the connection from the SOM population to the VIP population. The numerator is the gain of the pathway which connects these two populations via the PCs. When $J_{EE} > J_{EE}^*$ the negative contribution of the disinhibitory loop PC-VIP-SOM-PC dominates in the expression of $\chi_{II}$. It is the opposite when $J_{EE} < J_{EE}^*$. The stability of the balanced state provides other necessary conditions that the interactions must satisfy (see Materials and methods). In particular, the determinant of the interaction matrix, $J$, must be positive.

The difference between the behaviors in *Figure 4B and C* can now be understood as follows: in *Figure 4B*, $J_{EE} > J_{EE}^*$ and $\chi_{II} = 1.6 > 0$, thus, $r_I$ increases with $I_{opto}$; in *Figure 4C*, $J_{EE} < J_{EE}^*$ and $\chi_{II} = -5.1 < 0$ and thus, $r_I$ decreases. Remarkably, in both cases the activities of the PC and VIP populations normalized to baseline, are always equal (*Figure 4B–C*, right panel). This is a consequence of the balance of excitatory and inhibitory inputs into the SOM population which implies that $r_E$ and $r_V$ are proportional (see Materials and methods, *Equation 19.3*).

In *Figure 4B*, the activity of the SOM population decreases with the laser intensity. This also stems from the fact that $J_{EE} > J_{EE}^*$ (Appendix 1C, *Equations A31-34*). This qualitative behavior is therefore independent of parameter sets, provided that inequality (1) is satisfied. In contrast, for parameters for which $J_{EE} < J_{EE}^*$ the activity of the SOM population either decreases or increases with $I_{opto}$ depending on other parameters. Moreover, it is straightforward to prove that if $J_{EE} > J_{EE}^*$, the product $\chi_{EI}\chi_{IE}$ is positive (Appendix 1C). Since we assumed that $r_E$ decreases upon photostimulation of PV neurons, namely $\chi_{EI} < 0$, this implies that $\chi_{IE}$ is also negative. In other words, in Model 1, a non-paradoxical response of the PV population upon PV photostimulation implies that the PV activity *decreases* when PCs are photostimulated.

When $I_{opto}$ is sufficiently large, the solution of the four balance equations will contain one or more populations for which $r_\alpha < 0$. Obviously such a solution is inconsistent. Instead, other solutions should be considered where at least one population has a firing rate which is zero and the firing rates of the other populations is determined by a new system of linear equations with lower dimensions (see Materials and methods, Appendix 1C). Consistency requires that in these solutions the net input is hyperpolarizing for the populations with $r_\alpha = 0$. As a consequence, the network population activities are in general piecewise linear in $I_{opto}$ (*Figure 4—figure supplement 2*).

The large $N$, $K$ analysis provides precious insights into the dynamics of networks with reasonable size and connectivity. In particular, we will show that the criterion for the paradoxical effect, *Equation 1*, remains valid up to small corrections. Although it is possible to treat analytically the dependence of $r_\alpha$ on $I_{opto}$ for finite $K$, these calculations are very technical and beyond the scope of this paper. Instead here, we proceed with numerical simulations.

### Numerical simulations for $J_{EE} > J_{EE}^*$

*Figure 5* depicts the results of our numerical simulations of Model 1 for the same parameters as in *Figure 4B* (see Materials and methods, *Tables 3–4*). The response of PV neurons is non-paradoxical: the activity of the PV population increases monotonically with $\Gamma_{opto}$ in the whole range (*Figure 5A*). Concurrently, the population activities of PC, SOM and VIP neurons monotonically decrease with $\Gamma_{opto}$ (*Figure 5A-B*). For sufficiently large $\Gamma_{opto}$, PCs become very weakly active and the SOM and VIP populations dramatically reduce their firing rates. The variations with $\Gamma_{opto}$ of $r_E$, $r_I$, $r_S$ and $r_V$ and are robust to changes in the average connectivity, $K$ (*Figure 5—figure supplement 1*) and in qualitative agreement with the predictions of the large $N$, $K$ limit (*Figure 4B* Appendix 1C, *Figure 4—figure supplement 2*).

To test the robustness of our results with respect to changes in the interaction strengths, we generated 100 networks with $J_{\alpha\beta}$ chosen at random within a range of ±10% of those of *Figure 4B*. All the networks exhibited a balanced state which was stable with respect to slow rates fluctuations in the large $N$, $K$ limit. We simulated those networks with $K = 500$ and computed the population activity at baseline and for $\Gamma_{opto} = 0.07 mW.mm^{-2}$. For all these networks, the results were consistent with the one of the control set: for $\Gamma_{opto} = 0.07 mW.mm^{-2}$, $r_I$ was larger and $r_E$, $r_S$, $r_V$ were smaller than baseline (*Figure 5—figure supplement 2*). However, a small percentage of these networks (10%) exhibited oscillations with at most an amplitude 20% of their mean in the firing rates. Apart from that, the results were robust to changes in $J_{\alpha\beta}$.

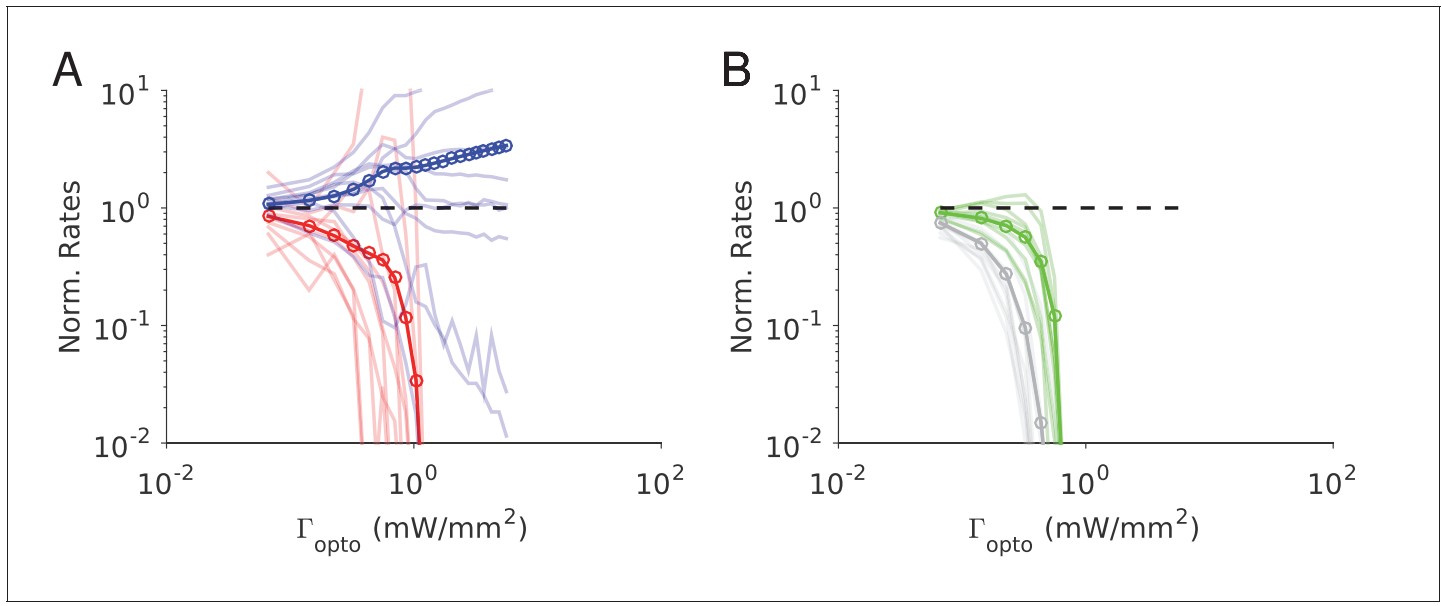

**Figure 5.** Numerical simulations of Model 1 for $J_{EE} > J_{EE}^*$. Responses of the neurons normalized to baseline *vs.* the intensity of the laser, $\Gamma_{opto}$. (**A**) Activities of PCs and PV neurons: the PV response is not paradoxical. (**B**) Activities of SOM and VIP neurons. Color code as in *Figure 4*. Thick lines: population averaged responses. Thin lines: responses of 10 neurons randomly chosen in each population. Firing rates were estimated over 100s. Parameters: $K = 500$, $N = 76800$. Other parameters as in *Tables 3–4*. The baseline activities are: $r_E = 3.3$ H$_z$, $r_I = 6.5$ H$_z$, $r_S = 5.9$ H$_z$, $r_V = 3.5$ H$_z$. The online version of this article includes the following figure supplement(s) for figure 5:

**Figure supplement 1.** Model 1 with $J_{EE} > J_{EE}^*$.
**Figure supplement 2.** Model 1 with $J_{EE} > J_{EE}^*$.
**Figure supplement 3.** Model 1 with $J_{EE} > J_{EE}^*$.

**Table 3.** Synaptic time constants.

| t$_{ab}$ (ms) | PC | PV | SOM | VIP |
|---|---|---|---|---|
| PC | 4 | 2 | 2 | N/A |
| PV | 2 | 2 | 4 | N/A |
| SOM | 2 | N/A | N/A | 4 |
| VIP | 4 | 2 | 4 | N/A |

In contrast to what happens in the large *N, K* limit (*Figure 4B*, right panel), in the results depicted in *Figure 5* the activity of the PC and VIP populations are not proportional. Moreover, in the large *K* limit, PC and VIP neurons are inactivated before the SOM population is. For *K* = 500, VIP is the first population to be silenced followed by the SOM and finally the PC population. Simulations with increasing values of *K* show that these differences are due to substantial finite *K* effects (*Figure 5—figure supplement 1*).

*Figure 5* also depicts the changes in the firing rates (normalized to baseline) with $\Gamma_{opto}$ for several example neurons. These changes are highly heterogeneous across neurons within each population. Whereas the population average varies monotonically, individual cells activity can either increase or decrease and the response can even be non-monotonic with $\Gamma_{opto}$.

The heterogeneity in the single neuronal responses are also clear in *Figure 6A–B* that plots, for two different light intensities, the perturbed firing rate *vs.* baseline for PCs and PV neurons. Remarkably, in both populations a significant fraction of neuron exhibits a response which is incongruous with the population average. The pie charts in *Figure 6* depict the fraction of PCs and PV neurons which increased, decreased, or did not change their firing rates. The fraction of neurons whose activity is almost completely suppressed, is also shown. Remarkably, even for $\Gamma_{opto} = 1.0mW.mm^{-2}$, some of the PCs show an activity increase. Moreover, the fraction of PV neurons whose firing rate increases is less for $\Gamma_{opto} = 1.0mW.mm^{-2}$ than $\Gamma_{opto} = 0.5mW.mm^{-2}$. It should be noted that in the model all PV neurons receive the same optogenetic input, therefore, the heterogeneity in the response is not due to whether or not the PV neurons were "infected". This heterogeneity is solely due to the randomness in the connectivity.

## Numerical simulations for $J_{EE}$<$J_{EE}^*$

*Figure 7* depicts the results of our numerical simulations of Model 1 when $J_{EE}$<$J_{EE}^*$. Parameters are the same as in *Figure 4C* (see Materials and methods, *Tables 3–5*). The population activities of PCs and VIP neurons, $r_E$ and $r_V$, decrease monotonically with the laser intensity, $\Gamma_{opto}$. Conversely, the variations of the activities of the PV and SOM populations, $r_I$ and $r_S$, are non-monotonic with $\Gamma_{opto}$. For small light intensities, $r_I$ decreases and then abruptly increases with larger $\Gamma_{opto}$; $r_S$ exhibits the opposite behavior. Remarkably, when $r_I$ is minimum, $r_S$ is maximum for nearly the same value of $\Gamma_{opto}$. We show in *Figure 7—figure supplement 1* that this proportional decrease only happens in a small region of parameter space when the determinant of the interaction matrix, $J_{\alpha\beta}\ \epsilon_\beta$, is close to zero.

This behavior is qualitatively similar to the one derived in the large *N, K* limit (*Figure 4—figure supplement 2*). As suggested by the large *N, K* analysis, the paradoxical response of the PV neurons in the simulations, is driven by the positive feedback loop PC-VIP-SOM-PC (*Figure 4—figure supplement 1*). Remarkably, when the activity of the PV neurons is minimum, the PCs are still substan-

**Table 4.** Connection strength matrix for $J_{EE}$>$J_{EE}^*$ (rows: postsynaptic populations; columns: presynaptic populations)

| $J_{\alpha\beta}$ (mA. ms.cm$^{-2}$) | Feedforward | PC | PV | SOM | VIP |
|---|---|---|---|---|---|
| PC | 34 | 20 | 26.4 | 41 | 0 |
| PV | 27 | 44 | 28 | 35.6 | 0 |
| SOM | 0 | 24 | 0 | 0 | 14 |
| VIP | 39 | 12 | 35.2 | 35 | 0 |

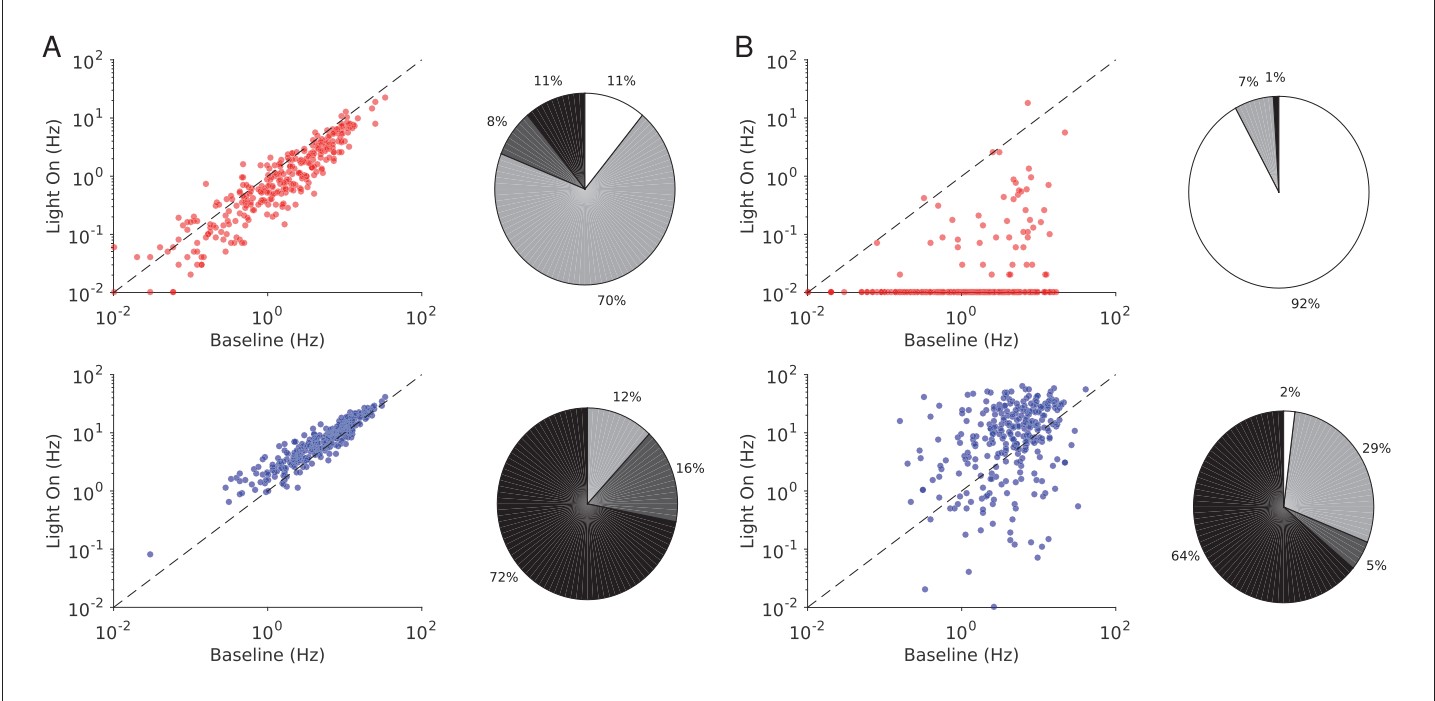

**Figure 6.** Single neuron firing rates in the PC and PV populations upon PV activation for two values of the light intensity (Model 1 with $J_{EE}>J_{EE}^*$). (A) Single neuron firing rates at baseline vs. at $\Gamma_{opto} = 0.5mW.mm^{-2}$. (B) Same for $\Gamma_{opto} = 1mW.mm^{-2}$. Top: PCs (red). Bottom: PV neurons (blue). Scatter plots of 300 randomly chosen PC and PV neurons. Pie charts for the whole population. The pie charts show the fraction of neurons which increase (black) or decrease (light gray) their activity compared to baseline. Dark gray: Fraction of neurons with relative change smaller than 0.1Hz. White: fraction of neurons with activity smaller than 0.1Hz upon PV photostimulation. Firing rates were estimated over 100s. Neurons with rates smaller than 0.01Hz are plotted at 0.01Hz. Parameters as in *Figure 5*.

tially active (40% of baseline level). This is due to finite $K$ corrections to the large $N$, $K$ predictions (*Figure 7—figure supplement 2*). These corrections are strong and scale as $\frac{1}{\sqrt{K}}$ (Appendix 1C). Indeed, even for $K$ as large as 2000, $r_E$ is still 25% of the baseline when $r_I$ is minimum.

We checked the robustness of these results with respect to changes in the interaction parameters as we did for $J_{EE}>J_{EE}^*$. We found that for small light intensity all the 100 simulated networks were operating in the balanced state and exhibited the paradoxical effect (*Figure 7—figure supplement 3*).

Finally, the single neuron responses are highly heterogeneous. *Figure 8* plots the perturbed activities of PCs and PV neurons vs. their baseline firing rates for two light intensities. In *Figure 8A*, the PV response is paradoxical. This is not the case in *Figure 8B*. Interestingly, the fraction of PV neurons incongruous with the population activity is larger for $\Gamma_{opto} = 0.5mW.mm^{-2}$ than for $\Gamma_{opto} = 1.0mW.mm^{-2}$. For both light intensities the activity of almost all the PCs is decreased.

## Four-population network: Model 2

In S1, in the range of laser intensities in which the PV response is paradoxical, the decrease of the PC and PV activity is proportional. This feature of the data can be accounted for in Model 1 but only with a fine-tuning of the interaction parameters (*Figure 7—figure supplement 1* and *Figure 7—figure supplement 4*). This prompted us to investigate whether a different architecture could account robustly for this remarkable property. Our hypothesis is that this property is a direct consequence of the balance of excitation and inhibition.

## Theory in the large $N, K$ limit

We first considered the three-population model depicted in *Figure 9A*. It consists of the PC, PV and SOM populations. SOM neurons receive strong inputs from PCs and PV neurons, but do not interact

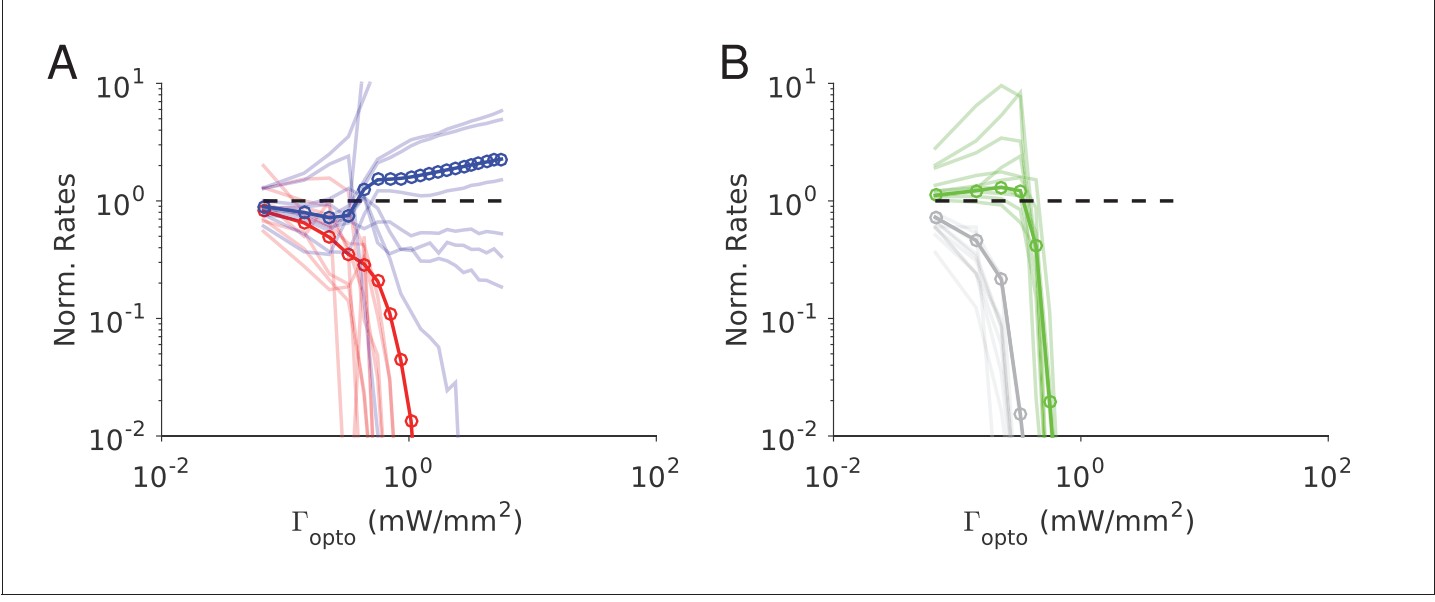

**Figure 7.** Numerical simulations of Model 1 for $J_{EE}<J^*_{EE}$. Responses of the neurons normalized to baseline *vs.* the intensity of the laser, $\Gamma_{opto}$. (**A**) Activities of PCs and PV neurons: the PV response is paradoxical. (**B**) Activities of SOM and VIP neurons. Color code as in **Figure 4**. Thick lines: population averaged responses. Thin lines: responses of 10 neurons in each population. Firing rates were estimated over 100s. Parameters: $K = 500$, $N = 76800$. Other parameters as in **Tables 3–5**. The baseline activities are: $r_E = 4.8$ Hz, $r_I = 11.2$ Hz, $r_S = 7.1$ Hz, $r_V = 5.3$ Hz. The online version of this article includes the following figure supplement(s) for figure 7:

**Figure supplement 1.** Model 1 for $J_{EE}<J^*_{EE}$.
**Figure supplement 2.** Model 1 with $J_{EE}>J^*_{EE}$.
**Figure supplement 3.** Model 1 with $J_{EE}<J^*_{EE}$.
**Figure supplement 4.** Model 1.
**Figure supplement 5.** Model 1 with $J_{EE}<J^*_{EE}$.

with each other and do not receive feedforward external inputs. In the large $N$, $K$ limit, the balance of excitation and inhibition of the SOM population reads (see Materials and methods, **Equation 20.2**).

$$J_{SE}r_E - J_{SI}r_I = 0 \tag{2}$$

Therefore, the activities of the PC and PV populations are always proportional. However, as we show in (Appendix 1D) a three-population network with such an architecture cannot exhibit the paradoxical effect.

We therefore considered a network model in which a third inhibitory population, referred to as 'X', is added without violating **Equation (3)** (**Figure 9B**). This requires that SOM neurons do not receive inputs from X neurons (Appendix 1D). This network exhibits the paradoxical effect if and only if $J_{SE}J_{EX}J_{XS}>J_{XX}J_{ES}J_{SE}$, that is if the gain of the positive feedback loop, SOM-X-PC-SOM, is sufficiently strong (Appendix 1D). Obviously, this condition simplifies and reads

**Table 5.** Connection strength matrix for $J_{EE}<J^*_{EE}$ (rows: postsynaptic populations; columns: presynaptic populations).

| $J_{\alpha\beta}$ (mA. ms.cm$^{-2}$) | Feedforward | PC | PV | SOM | VIP |
|---|---|---|---|---|---|
| PC | 52 | 17.4 | 34.4 | 32.8 | 0 |
| PV | 39 | 36.6 | 29.2 | 28.8 | 0 |
| SOM | 0 | 24.2 | 0 | 0 | 16.8 |
| VIP | 30 | 31.2 | 31 | 14.6 | 0 |

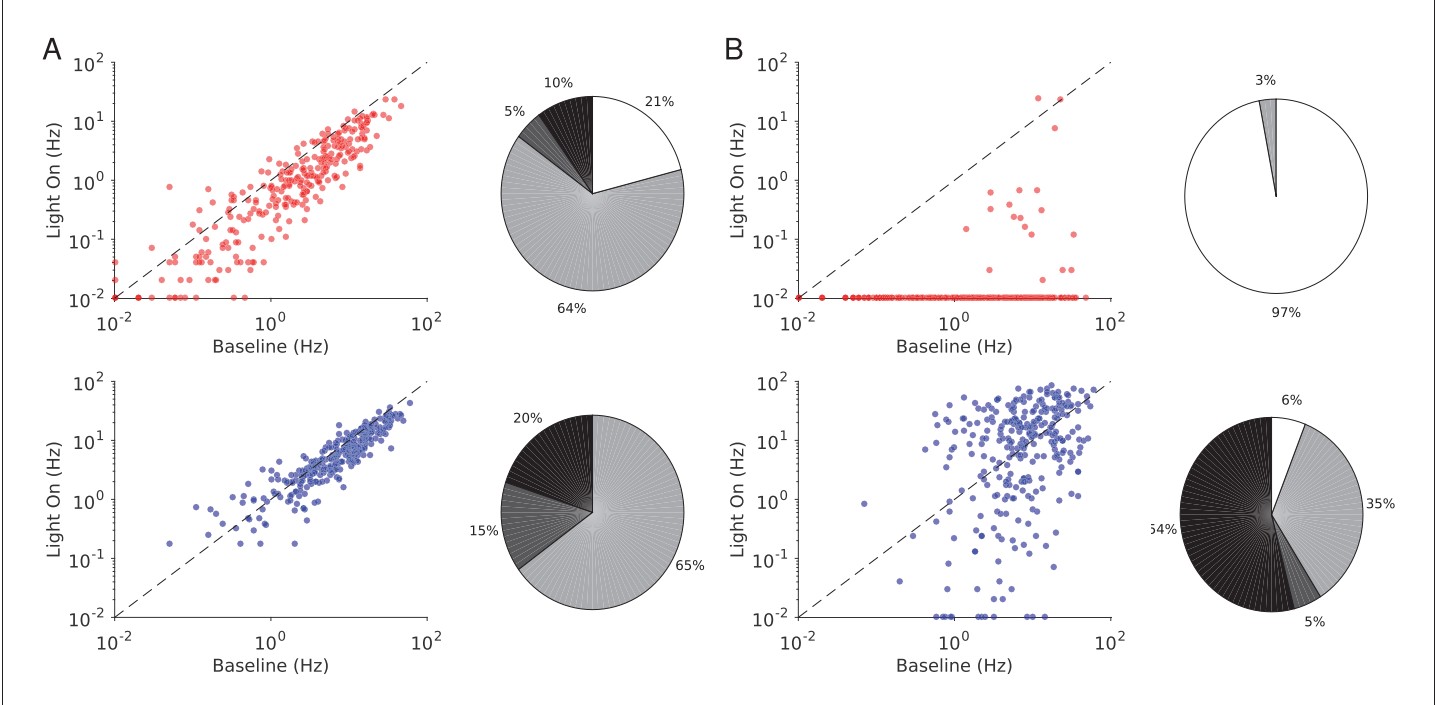

**Figure 8.** Single neuron firing rates in the PC and PV populations upon PV activation for two values of the light intensity (Model 1 with $J_{EE}<J_{EE}^*$). (A) Single neuron firing rates at baseline *vs.* at $\Gamma_{opto} = 0.5 mW.mm^{-2}$. (B) Same for $\Gamma_{opto} = 1mW.mm^{-2}$. Top: PCs. Bottom: PV neurons. Scatter plots of 300 randomly chosen PC and PV neurons. Pie charts for the whole population. Firing rates were estimated over 100s simulation time. Neurons with rates smaller than 0.01Hz are plotted at 0.01Hz. Color code as in **Figure 6**. Parameters as in **Figure 7**.

$$J_{EX}J_{XS}>J_{XX}J_{ES} \tag{3}$$

Remarkably, this inequality does not depend on $J_{EE}$. This is in contrast to what happens in Model 1 where the paradoxical effect occurs only if $J_{EE}$ is small enough (see **Equation (2)**).

As in Model 1, we further required that the activity of the PC population increases with its feed-forward external input. This adds the constraint (Appendix 1D):

$$J_{IX}J_{XS}>J_{XX}J_{IS} \tag{4}$$

**Equations (3-5)** do not depend on $J_{XI}$. For simplicity, we take $J_{XI} =0$ and refer to the resulting architecture as Model 2.

In **Figure 9C**, the slope of the PV population activity changes from negative to positive while PCs are still active. This is because if SOM neurons are completely suppressed, the loop SOM-X-PC-SOM which is responsible for the paradoxical effect, is not effective anymore. Interestingly, the analytical calculations also show that, when the SOM population activity vanishes, the activity of the X population is maximum. Since the SOM population is inactive before PCs, there is a range of laser intensities where the activity of the latter keeps decreasing while the activity of the PV population increases. Once PCs are inactive, the activity of the X population do not vary with $I_{opto}$. This is because then they only receive a constant feedforward excitation from outside the network which is balanced by their strong recurrent mutual coupling, $J_{XX}$.

## Simulations for finite K

These features are also observed in our simulations depicted in **Figure 10**. For small laser intensities, the network exhibits a paradoxical effect where the activities of the PC and PV populations decrease with $\Gamma_{opto}$ and in a proportional manner (**Figure 10A**), until the SOM neurons become virtually inactive (**Figure 10B**). At that value, $r_I$ is minimum and $r_X$ is maximum. For larger $\Gamma_{opto}$, $r_I$ increases while $r_E$ keeps decreasing and is still substantial. After $r_E$ has vanished, $r_X$ saturates but $r_I$ continues to

increase. All these results are robust to changes in the connectivity, K (*Figure 10—figure supplement 1*) as well as to changes in the interaction parameters (*Figure 10—figure supplement 2*). Single neuron responses are more heterogeneous than in the experimental data (*Figure 11*). It should be noted however that we did not tune parameters to match the experimental heterogeneity.

## Discussion

We studied the response of cortex to optogenetic stimulation of parvalbumin positive (PV) neurons and provided a mechanistic account for it. We photostimulated the PV interneurons in layer 2/3 and layer 5 of the mouse anterior motor cortex (ALM). In layer 2/3 photostimulation increased PV activity and decreased the response of the PCs on average. In contrast, in layer five the response of the PV population was paradoxical: *both* PC and PV activity decreased on average. This is similar to what we found in the mouse somatosensory cortex (S1) (*Li et al., 2019*). To account for these results, we first investigated the dynamics of networks of one excitatory and one inhibitory population of spiking neurons. We showed that two-population network models of strongly interacting neurons do not fully account for the experimental data. This prompted us to investigate the dynamics of networks consisting of more than one inhibitory population.

We considered two network models both consisting of one excitatory and three inhibitory populations. Interneurons are known to be unevenly distributed throughout the cortex. For instance, SOM neurons have been reported to be most prominent in layer five whereas VIP neurons are mostly found in layer 2/3 (*Tremblay et al., 2016*). Instead of giving a complete description of these layers and all neuronal populations they include, we propose here models with the minimal number of inhibitory populations that can account for the data.

The three inhibitory populations in Model 1 represent PV, somatostatin positive (SOM) and vasoactive intestinal peptide (VIP) interneurons with a connectivity similar to the one reported in primary visual cortex (*Pfeffer et al., 2013*) and S1 layer 2/3 (*Lee et al., 2013*). In Model 2, the first two inhibitory populations likewise represent PV and SOM neurons and the third population, denoted as X, represents an unidentified inhibitory subtype. The main difference with Model one is that here, the third population does not project to SOM neurons.

Depending on network parameters, the response of PV neurons in Model one can be paradoxical or not. To have equal relative suppression of the PCs and PV activities, however, interaction parame-

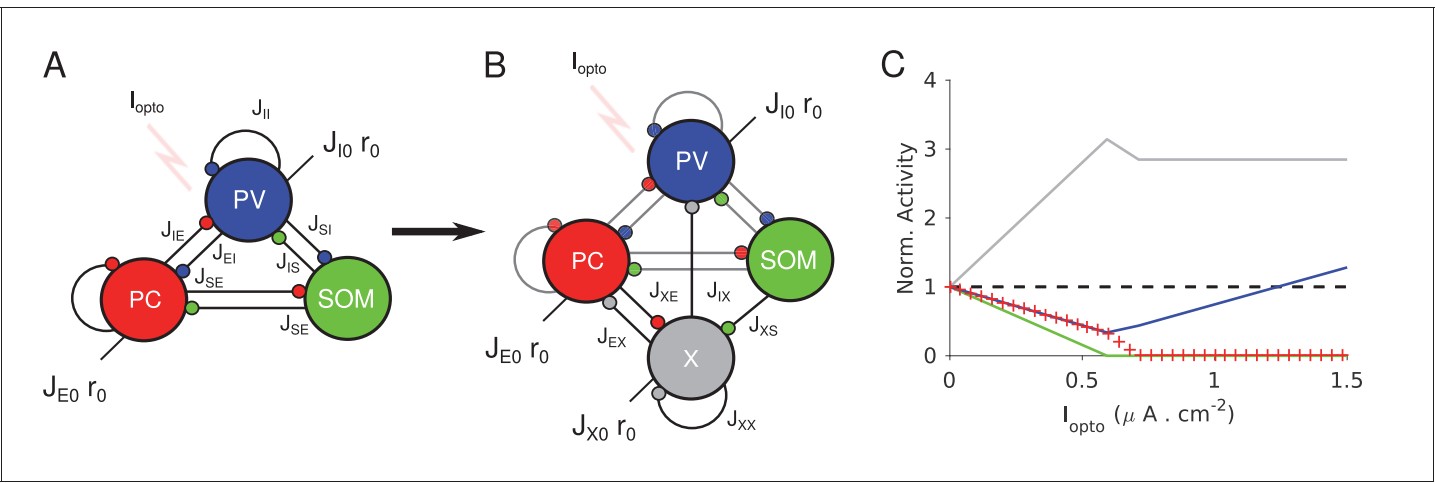

**Figure 9.** Network models with proportional change in the PC and PV activities upon photostimulation of the PV population. (**A**) A three-population network consisting of PCs, PV and SOM neurons. SOM neurons only receive projections from the PC and PV populations. (**B**) Model 2 consists of four populations: PC, PV, SOM and an unidentified inhibitory population, X. The population X projects to the PC, the PV population and to itself. The PC population projects to X. (**C**) Population activities normalized to baseline *vs.* $I_{opto}$ in the large $N$, $K$ limit. PC and PV populations decrease their activity with $I_{opto}$ in a proportional manner. Parameters as in *Tables 6–7*. Baseline firing rates are: $r_E$ = 3.0 Hz, $r_I$ = 6.7 Hz, $r_S$ = 6.4 Hz, $r_X$ = 3.8 Hz. The online version of this article includes the following figure supplement(s) for figure 9:

**Figure supplement 1.** Model 2.

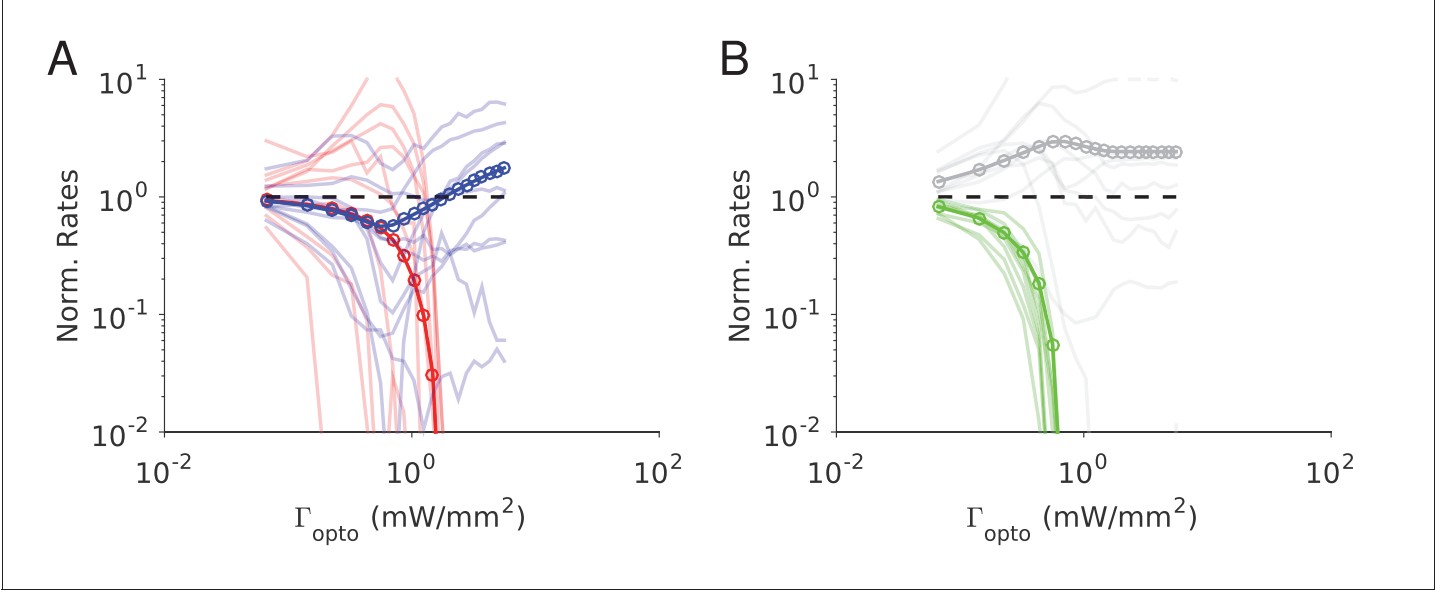

**Figure 10.** Numerical simulations of Model 2. Responses of the neurons normalized to baseline *vs.* the intensity of the laser, $\Gamma_{opto}$. (A) Activities of PCs and PV neurons: for small $\Gamma_{opto}$, the PV response is paradoxical and the suppression of the PC and PV population activities relative to baseline are the same. (B) Activities of SOM and X neurons. Color code as in *Figure 9*. Thick lines: population averaged responses. Thin lines: responses of 10 neurons randomly chosen in each population. Firing rates were estimated over 100s. Parameters: $K = 500$, $N = 76800$. Other parameters as in *Tables 6–7*. The baseline activities are: $r_E = 4.2$ Hz, $r_I = 6.8$ Hz $r_S = 7.0$ Hz, $r_X = 3.9$ Hz.

The online version of this article includes the following figure supplement(s) for figure 10:

**Figure supplement 1.** Model 2.
**Figure supplement 2.** Model 2.
**Figure supplement 3.** Model 2.

ters have to be fine-tuned. In Model 2, the relative changes in the PC and PV activity are the same independent of interaction parameters.

For a two-population network, the paradoxical effect only occurs when it is inhibition stabilized (*Pehlevan and Sompolinsky, 2014*; *Wolf et al., 2014*). This is because the mechanism requires strong recurrent excitation. In the four-population networks we studied, however, the mechanism responsible for paradoxical effect is different. It involves a disinhibitory loop. In fact, strong recurrent excitation prevents the paradoxical effect in these networks. Therefore, the observation of the paradoxical effect upon PV photo-excitation is not a proof that the network operates in the ISN regime.

## Strong vs. weak interactions

Cortical networks consist of a large number ($N$) of neurons each receiving a large number of inputs ($K$). Because $N$ and $K$ are large, one expects that a network behaves similar to a network where $N$ and $K$ are infinite. In this limit the analysis is simplified and the mechanisms underlying the dynamics are highlighted. When taking the large $K$ limit one needs to decide how the interaction strengths scale with $K$. Two canonical scalings can be used: in one the interactions scale as $1/K$ (*Hansel and Sompolinsky, 1992*; *Hennequin et al., 2018*; *Knight, 1972*; *Rubin et al., 2015*), in the other as $1/\sqrt{K}$ (*Darshan et al., 2017*; *Renart et al., 2010*; *Rosenbaum et al., 2017*; *van Vreeswijk and Sompolinsky, 1996*). These differ in the strength of the interactions. For instance, for $K = 900$ interactions are weaker by a factor 30 in the first scaling than in the second. Importantly, these two scalings give rise to qualitatively different dynamical regimes.

When interactions are strong, the excitatory and inhibitory inputs are both very large (of the order of $K \cdot \frac{1}{\sqrt{K}} = 1$). They, however, dynamically *balance* so that the temporal average of the net input and its spatial and temporal fluctuations are comparable to the rheobase (*Van Vreeswijk and Sompolinsky, 2005*; *van Vreeswijk and Sompolinsky, 1998*), Appendix 1A. In this *balanced regime*, the average firing rates of the populations are determined by a set of linear equations: the "balance

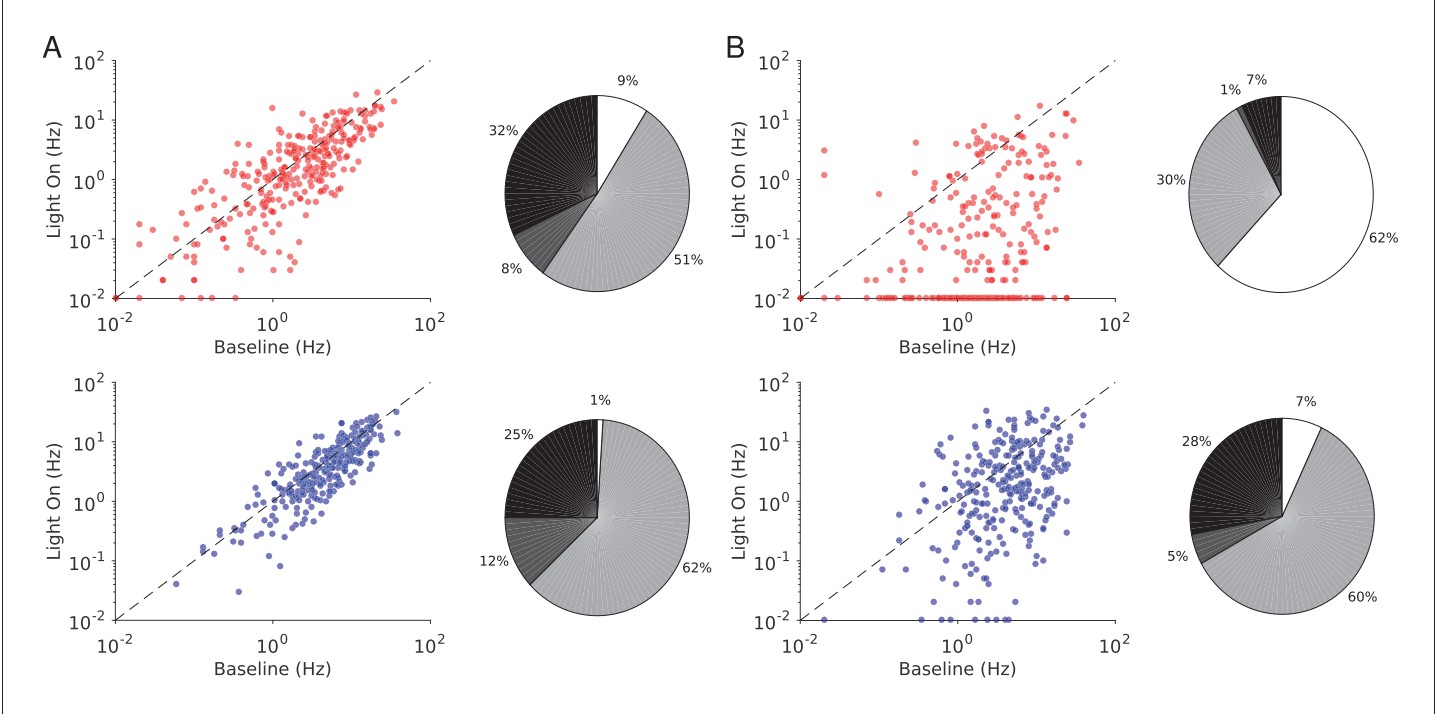

**Figure 11.** Single neuron firing rates in the PC and PV populations upon PV activation for two values of the light intensity (Model 2). (**A**) Single neuron firing rates at baseline *vs.* at $\Gamma_{opto} = 0.5mW.mm^{-2}$. (**B**) Same for $\Gamma_{opto} = 1mW.mm^{-2}$. Top: PCs. Bottom: PV neurons. Scatter plots of 300 randomly chosen PC and PV neurons. Pie charts for the whole population. Firing rates were estimated over 100s. Neurons with rates smaller than 0.01Hz are plotted at 0.01Hz. Color code as in *Figure 6*. Parameters as in *Figure 10*.

equations". These do not depend on the neuronal transfer function. For large but finite *K*, the network operates in an approximately balanced regime. In this regime, the population activities are well approximated by the balance equations, interspike intervals are highly irregular and firing rates are heterogeneous across neurons.

When the interactions are weak, excitatory and inhibitory inputs are both comparable to the rheobase even when *K* is large, but their spatial and temporal fluctuations vanish as *K* increases. The activity of the network is determined by a set of coupled non-linear equations which depends on the neuronal transfer function. For large but finite *K*, the firing of the neurons is weakly irregular and heterogeneities mostly arise from differences in the intrinsic properties of the neurons.

In which of these regimes does cortex operate *in-vivo*? This may depend on the cortical area and on whether the neuronal activity is spontaneous or driven (*e.g.* sensory, associative, or motor related). There are, however, several facts indicating that the approximate balanced regime may be ubiquitous. Many cortical areas exhibit highly irregular spiking (*Shinomoto et al., 2009*) and heterogeneous firing rates (*Hromádka et al., 2008*; *Roxin et al., 2011*). Excitatory and inhibitory postsynaptic potentials (PSPs) are typically of the order of 0.2 to 2mV or larger (*Levy and Reyes, 2012*; *Ma et al., 2012*; *Pala and Petersen, 2015*; *Seeman et al., 2018*). Model networks with PSPs of

**Table 6.** Default parameters of Model 2.
Synaptic time constants in Model 2.

| $t_{ab}$ **(ms)** | **PC** | **PV** | **SOM** | **X** |
|---|---|---|---|---|
| PC | 4 | 2 | 2 | 4 |
| PV | 2 | 2 | 4 | 4 |
| SOM | 2 | 2 | N/A | N/A |
| X | 2 | N/A | 4 | 2 |

**Table 7.** Connection strength matrix (rows: postsynaptic populations; columns: presynaptic populations).

| $J_{\alpha\beta}$ (mA ms.cm$^{-2}$) | Feedforward | PC | PV | SOM | VIP |
|---|---|---|---|---|---|
| PC | 48 | 20 | 30 | 32 | 36 |
| PV | 29 | 40 | 28 | 16 | 32 |
| SOM | 0 | 26 | 12 | 0 | 0 |
| VIP | 24 | 24 | 0 | 36 | 22 |

these sizes and reasonable number of neurons and connections exhibit all the hallmarks of the balanced regime (*Amit and Brunel, 1997*; *Hansel and Mato, 2013*; *Hansel and van Vreeswijk, 2012*; *Lerchner et al., 2006*; *Pehlevan and Sompolinsky, 2014*; *Argaman and Golomb, 2018*; *Rao et al., 2019*; *Roudi and Latham, 2007*; *Roxin et al., 2011 Van Vreeswijk and Sompolinsky, 2005*). Moreover, there is experimental evidence of co-variation of excitatory and inhibitory inputs into cortical neurons (*Haider et al., 2006*; *Shu et al., 2003*). Finally, in cortical cultures synaptic strengths have been shown to approximately scale as $1/\sqrt{K}$ (*Barral and D Reyes, 2016*). Therefore in this paper we focused on cortical network models in which interactions are strong, that is of the order of $1/\sqrt{K}$.

## Model 1 accounts for the responses in ALM layer 2/3 and layer 5

In Model 1, whether the network exhibits a paradoxical effect depends on the value of the ratio $\rho = J_{EE}/J_{EE}^*$ where $J_{EE}^* \equiv J_{VE}J_{ES}/J_{VS}$. Here, $J_{\alpha\beta}$, $\alpha, \beta \in \{E, S, V\}$, is the strength of the connection from population β to population $\alpha$. When ρ > 1, the PV response is non-paradoxical and its activity increase can be substantial well before suppression of the PC activity. On the other hand when ρ > 1, the PV response is paradoxical and the PV activity reaches its minimum for light intensities at which the PCs are still substantially active.

In ALM layer 2/3, the activity of the PV population increases with the light intensity while the activity of the PC decreases on average. Remarkably, our experiments showed that the increase in the PV activity was already substantial for small light intensities, where the PCs were still significantly active. In ALM layer 5 the activity of the PV population initially decreased with the light intensity together with the activity of the PC population. As the light intensity is further increased, the PV activity reaches a minimum after which it increases. At this minimum, the PC activity is still substantial.

Thus, Model 1 accounts for our experimental findings in ALM layer 2/3 provided that $J_{EE}$ is sufficiently large. It accounts for the paradoxical effect in layer 5 provided that $J_{EE}$ is sufficiently small. Note that this does not mean that $J_{EE}$, is larger in the former layer as compared to the latter. The interactions $J_{VE}$, $J_{ES}$ and $J_{VS}$ are likely to be layer dependent (*Jiang et al., 2015*) and therefore so is the value of $J_{EE}^*$.

## Model 2 accounts for the paradoxical effect in S1 while model 1 would require fine-tuning

Similar to ALM layer 5, the PV response in S1 is paradoxical. Remarkably however, in S1 the relative suppression of the PC and PV activities is the same for low light intensity. Model 1 can account for this feature only when the interaction parameters are fine-tuned. In contrast, in Model 2 the co-modulation of the PC and PV activities stems from the architecture and therefore occurs in a robust manner. Furthermore, it can equally well account for the fact that in S1 the PV activity reaches its minimum when the PC population is active.

Note that in ALM layer 5 the difference between the slopes of the PC and PV population activities is not significantly different (p>0.05). Therefore, we cannot exclude that Model 2 describes ALM layer 5.

The main difference between Models 1 and 2 is that in Model 1, the third inhibitory population (VIP) projects to SOM neurons while in Model 2, the third population (X) does not. This suggests that population X is not the VIP population. For example, X could be chandelier cells that do not express the PV marker (*Jiang et al., 2015*) Alternatively, population X could describe the *effective* interaction of several inhibitory populations with PC and PV neurons.

## Models 1 and 2 account for the heterogeneity of single neuron responses

The responses of PCs and PV neurons in the experimental data are highly heterogeneous across cells. Indeed in ALM layer 5 and S1, PV neurons on average show a paradoxical response but at the single neuron level the effect of the laser stimulation is very diverse. Moreover, the firing rate of a neuron can vary monotonically or non-monotonically with the laser intensity. For instance, when stimulated, the firing rates of many PV neurons increase, although, on average the activity is substantially smaller than baseline. Conversely, for some PV neurons the paradoxical effect is so strong that the laser completely suppresses their activity.

We observed an even larger diversity in single neuron responses in our simulations of Model 1 and 2. We should emphasize that in the simulated networks all the neurons were identical and the cells in the same population received the same feedforward constant external input. The only possible source of heterogeneity therefore comes from the randomness in the network connectivity. The effect of this randomness on the network recurrent dynamics is however non-trivial: one may think that the effect of the fluctuations in the number of connections from neuron to neuron should average out since in the models the number of recurrent inputs per neuron is large ($K = 500$ or more). This is not what happens because in our simulations populations which are active operate in the balanced excitation/inhibition regime (*Roxin et al., 2011*; *van Vreeswijk and Sompolinsky, 1998*; *van Vreeswijk and Sompolinsky, 1996*). In this state, relatively small homogeneity in the number of connections per neuron is amplified to a substantial inhomogeneity in the response. Thus, strong heterogeneity in the response of neurons is not a prima facie evidence for the heterogeneity of the level of Channelrhodopsin expression in the cells nor is it for the diversity of the single neuron intrinsic properties.

## Limitations

We give here a qualitative account for the mechanisms underlying the responses of different cortical areas to optical stimulation. A *quantitative* analysis of the data, in particular of the heterogeneity is beyond our scope. Such an analysis would require a much larger number of PV neurons. Moreover, it would necessitate the use of more complicated neuronal models making the mathematical analysis intractable, limiting the investigation to simulations only and thus obscuring the mechanisms.

In our experiments, we expressed ReaChR in all PV neurons and in all layers in ALM. In particular, all PV neurons in layer 2/3 and layer five were simultaneously affected by the photostimulus. PCs in layer 2/3 project to layer 5 and receive feedback from the latter (*Hooks et al., 2013*; *Naka and Adesnik, 2016*). Interlaminar interactions are likely to also contribute to the effect of the photostimulation.

In our models, we did not take into account such interactions. Including strong connections from layer 2/3 PCs to neurons in layer 5 and/or feedback connections from layer 5 neurons to layer 2/3, could alter our interpretations. In the absence of data that reveal the nature of interlaminar interactions, extending our model to incorporate these is impractical given the large number of parameters to vary. Experiments in ALM and S1 where the optogenetic marker is expressed in only one layer at a time would constraint models which include interlaminar interactions and facilitate their analysis (*Moore et al., 2018*).

There is a large amount of experimental evidence indicating that different synapses can exhibit diverse dynamics depending on their pre and postsynaptic populations (*Ma et al., 2012*). For instance, recent studies have shown that PCs to PV synapses are depressing while the PCs to SOM synapses are highly facilitating (*Karnani et al., 2016*; *Xu et al., 2013*). Synaptic facilitation and depression mechanisms could give rise to dynamics which will make the network responses depend on the duration of the photostimulation. Here, we did not take into account short term plasticity. Mice neocortex mostly comprises PV, SOM and 5HT3aR expressing interneurons. There is a growing amount of experimental evidence indicating that these populations include different subtypes which may have distinct connectivity patterns (*Naka and Adesnik, 2016*; *Nigro et al., 2018*; *Tremblay et al., 2016*). In the present work, we only considered three populations of identical interneurons: PV, SOM and VIP or X. As the number of populations increases, the number of interaction parameters increases quadratically, making it a great challenge to uncover even simple mechanisms that could underlie the network responses.

## Comparison with previous theoretical work

The paradoxical effect was first described in *Tsodyks et al. (1997)* and *Ozeki et al. (2009)* for weak interactions using coarse grained two-population rate models (*Wilson and Cowan, 1972*). These models were extended in *Rubin et al. (2015)* to a spatially structured network to explain center-sur-round interactions and other contextual effects in primary visual cortex. They found that these effects can be accounted for if the neuronal transfer function is supralinear and the network is oper-ating in the inhibition stabilized regime (ISN). With supralinear transfer functions, whether or not the network exhibits a paradoxical effect depends on the background rate of the inhibitory neurons. These models were further extended by *Litwin-Kumar et al. (2016)* to networks consisting of PC, PV, SOM and VIP neurons with an architecture similar to *Pfeffer et al. (2013)*. They studied the effect of photostimulation of the different inhibitory populations on the responses and orientation tuning properties of the neurons. In a recent study (*Sadeh et al., 2017*) have investigated the effects of partial activation of PV neurons upon photostimulation in an ISN. They argued that depending on the degree of viral expression, the average response of the infected neurons can decrease or increase with the light intensity: it decreases only if a large proportion of the population is infected. (*Garcia Del Molino et al., 2017*) showed that due to the non-linearity in the neuronal transfer func-tion, the response of the network to stimulation can be different for different background rates. In particular, they showed that it can reverse the response of SOM neurons to VIP stimulation.

All these works considered inhibition stabilized networks in which the *total* recurrent excitation is so strong that the activity would blow up in the absence of inhibitory feedback. With our notations, this means that $G_E j_{EE} > 1/K$, where $G_E$ is the gain of the noise average transfer function (f-I curve) of the excitatory neurons. In fact, in these models all the interactions $j_{\alpha\beta}$ are of order $1/K$ so they are weak in our sense. Moreover, these studies considered networks that are so small that it is impossi-ble to extrapolate their results to mouse cortex size networks. Here we studied large network mod-els (N = 76800) with strong interactions, that is $j_{\alpha\beta}$ are of order $1/\sqrt{K}$ operating in the balanced regime. Note that such networks are ISNs provided that $j_{EE} \neq 0$. We showed that paradoxical effect can be present or not depending on the interaction parameters.

Since we used static synapses, changes in the background rates cannot reverse the paradoxical effect in our models. This is because with static synapses the balance equations are linear. One can recover this reversal if one introduces short-term plasticity which will make the balance equations nonlinear. We did not consider partial expression of channelrhodopsin in the PV population because our goal was to account for experimental data where virtually all neurons were infected. These effects have been studied in *Gutnisky et al. (2017)*; *Sanzeni et al. (2019)* in strongly coupled net-works of two populations yielding to the same conclusions as (*Sadeh et al., 2017*).

## Predictions

Our theory (Model 1) predicts that in ALM layer 2/3 the activity of the SOM and VIP populations will decrease upon PV photostimulation (*Figure 4B*). It also predicts that upon PC photoinhibition, the PV activity will increase whereas the activity of the SOM and VIP populations will decrease (*Figure 12A*). This is because in Model 1 when the PV response is non-paradoxical ($\chi_{II} > 0$) the prod-uct $X_{EI} X_{IE}$ is also positive (see Appendix 1C). Furthermore, in ALM layer 2/3 the population activity of PCs decreases upon PV photostimulation, $X_{EI} < 0$. Hence, $X_{IE}$ is negative. The balance of the PC and the VIP inputs into SOM neurons implies that VIP and PC activity covary. Finally, in Appendix 1C we show that if $X_{EE} > 0$ and $X_{IE} < 0$ then necessarily $X_{SE} > 0$. Thus, in ALM layer 2/3, the SOM popu-lation activity should decrease upon PC photoinhibition (*Figure 12A*).

In auditory and prefrontal cortex (*Pi et al., 2013*) as well as in S1 (*Lee et al., 2013*), photostimula-tion of VIP neurons, activates them ($X_{VV} > 0$) and disinhibits the PCs ($X_{EV} > 0$) through an inhibition of the SOM population ($X_{SV} > 0$). If this is also true in ALM layer 2/3, our model predicts that photo-stimulation of VIP neurons should increase the PV activity ($X_{IV} > 0$) (Appendix 1C, *Figure 12B*).

In S1 our theory (Model 2) predicts that the PC and PV activities will proportionally decrease upon PC photoinhibition (*Equation (3)*, Appendix 1D, *Figure 12C*). Photostimulation of the SOM neurons modifies *Equation (3)* and consequently, the changes in PC and PV activity no longer covary (*Figure 12D*). Thus, our theory can be tested by photostimulating PV neurons as in our experiment, while also photostimulating SOM neurons with a second laser with constant power. In this case, the

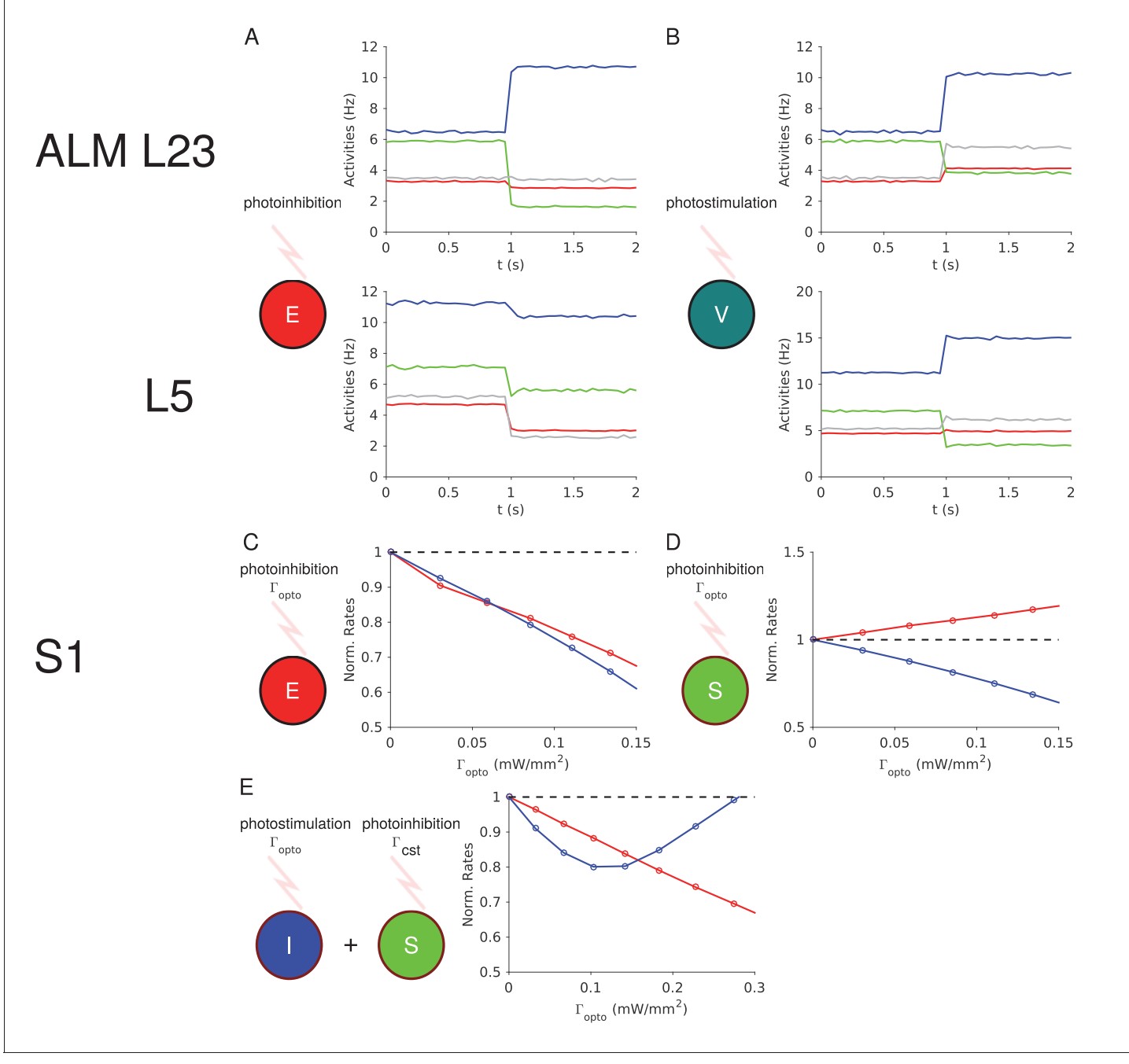

**Figure 12.** Predictions of the theory. (**A**) In ALM layer 2/3, the activity of the PV population decreases upon photoinhibition of the PCs. (**B**) In ALM layer 2/3, photostimulation of VIP neurons increases the activity of the PV population. (**C**) In S1, PV and PC activity decrease proportionally upon photoinhibition of the latter. (**D**) In S1, the PC and PV responses are not proportional upon photoinhibition of the SOM population. (**E**) In S1, upon photostimulation of PV neurons and photoinhibition of the SOM population with a constant input, the PV response is paradoxical but PC and PV responses are no longer proportional.

model predicts that S1 will still exhibit the paradoxical effect but that the responses of the PC and PV populations will no longer be proportional (*Figure 12E*).

## Perspectives

We only considered response of the neurons for a large radius of the laser beam. In a recent study *Li et al. (2019)*, have investigated the spatial profile of the response and its dependence on the light intensity. Our theory can be extended to incorporate spatial dependencies. Studying the interplay between the connectivity pattern and laser beam width in the response profile of the networks will provide further constraints on cortical architectures.

Due to the strong interactions in our models, the nonlinearity of the *single* neuron f-I curves hardly affects the population average responses. However, it influences the response heterogeneity that naturally arises in our theory (*Figures 6–8*). An alternative model for the paradoxical effect is the supralinear stabilized network (SSN) (*Rubin et al., 2015*) which relies on an expansive non-linearity of the input-output transfer function of the inhibitory *populations*. Whether this mechanism can account for our experimental data is an issue for further study. In particular, it would be interesting to know whether the SSN scenario can account for the strong heterogeneity in the responses and for the proportionality of the PC and PV population activities in S1. Answering these questions may provide a way to discriminate between the balance network and SSN theory.

# Materials and methods

### Key resources table

| Reagent type (species) or resource | Designation | Source or reference | Identifiers | Additional information |
|---|---|---|---|---|
| Genetic reagent (*Mus musculus*) | *Pvalb*-Ires-Cre | The Jackson Laboratory | JAX #008069 | |
| Genetic reagent (*Mus musculus*) | R26-CAG-LSL-ReaChR-mCitrine | The Jackson Laboratory | JAX #026294 | |

## Animals and surgery

The experimental data are from 9 PV-Ires-Cre x R26-CAG-LSL-ReaChR-mCitrine mice (age >P60, both male and female mice) (*Hooks et al., 2015*). three mice were used for photoinhibition in somatosensory cortex (S1). six mice were used for photoinhibition in anterior lateral motor cortex (ALM). All procedures were in accordance with protocols approved by the Janelia Research Campus and Baylor College of Medicine Institutional Animal Care and Use Committee.

Mice were prepared for photostimulation and electrophysiology with a clear-skull cap and a head-post (*Guo et al., 2014a*; *Guo et al., 2014b*). The scalp and periosteum over the dorsal surface of the skull were removed. A layer of cyanoacrylate adhesive (Krazy glue, Elmer's Products Inc) was directly applied to the intact skull. A custom made headbar was placed on the skull (approximately over visual cortex) and cemented in place with clear dental acrylic (Lang Dental Jet Repair Acrylic; Part# 1223-clear). A thin layer of clear dental acrylic was applied over the cyanoacrylate adhesive covering the entire exposed skull, followed by a thin layer of clear nail polish (Electron Microscopy Sciences, Part# 72180).

## Photostimulation

Light from a 594 nm laser (Cobolt Inc, Colbolt Mambo 100) was controlled by an acousto-optical modulator (AOM; MTS110-A3-VIS, Quanta Tech; extinction ratio 1:2000; 1μs rise time) and a shutter (Vincent Associates), coupled to a 2D scanning galvo system (GVA002, Thorlabs), then focused onto the brain surface (*Guo et al., 2014a*). The laser at the brain surface had a diameter of 2 mm. We tested photoinhibition in barrel cortex (bregma posterior 0.5 mm, 3.5 mm lateral) and ALM (bregma anterior 2.5 mm, 1.5 mm lateral).

To prevent the mice from detecting the photostimulus, a 'masking flash' pulse train (40 1 ms pulses at 10 Hz) was delivered using a LED driver (Mightex, SLA-1200–2) and 590 nm LEDs (Luxeon Star) positioned near the eyes of the mice. The masking flash began before the photostimulus started and continued through the end of the epoch in which photostimulation could occur.

The photostimulus had a near sinusoidal temporal profile (40 Hz) with a linear attenuation in intensity over the last 100–200 ms (duration: 1.3 s including the ramp). The photostimulation was

delivered at ~7 s intervals. The power (0.5, 1.2, 2.2, 5, 12 mW for S1 photostimulation; 0.3, 0.5, 1, 1.5, 2, 3.3, 5, 8, 15 mW for ALM photostimulation) were chosen randomly. Because we used a time-varying photostimulus, the power values reported here reflect the time-average.

## Electrophysiology

All recordings were carried out while the mice were awake but not engaged in any behavior. Extracellular spiking activity was recorded using silicon probes. We used 32-channel NeuroNexus silicon probes (A4 × 8–5 mm-100-200-177) or 64-channel Cambridge NeuroTech silicon probes (H2 acute probe, 25 µm spacing, two shanks). The 32-channel voltage signals were multiplexed, digitized by a PCI6133 board at 400 kHz (National Instruments) at 14 bit, demultiplexed (sampling at 25,000 Hz) and stored for offline analysis. The 64-channel voltage signals were amplified and digitized on an Intan RHD2164 64-Channel Amplifier Board (Intan Technology) at 16 bit, recorded on an Intan RHD2000-Series Amplifier Evaluation System (sampling at 20,000 Hz) using Open-Source RHD2000 Interface Software from Intan Technology (version 1.5.2), and stored for offline analysis.

A 1 mm diameter craniotomy was made over the recording site. The position of the craniotomy was guided by stereotactic coordinates for recordings in ALM (bregma anterior 2.5 mm, 1.5 mm lateral) or barrel cortex (bregma posterior 0.5 mm, 3.5 mm lateral).

Prior to each recording session, the tips of the silicon probe were brushed with DiI in ethanol solution and allowed to dry. The surface of the craniotomy was kept moist with saline. The silicon probe was positioned on the surface of the cortex and advanced manually into the brain at ~3 µm/s, normal to the pial surface. The electrode depth was inferred from manipulator depth and verified with histology. For ALM recordings, putative layer 2/3 units were above 450 µm and putative layer 5 units were below 450 µm (*Hooks et al., 2013*). For S1, our recording did not distinguish layers.

## Data analysis

The extracellular recording traces were band-pass filtered (300–6 kHz). Events that exceed an amplitude threshold (four standard deviations of the background) were subjected to manual spike sorting to extract single units (*Guo et al., 2014a*).

Our final data set comprised of 204 single units (S1, 95; ALM, 109). For each unit, its spike width was computed as the trough to peak interval in the mean spike waveform (*Guo et al., 2014a*). We defined units with spike width <0.35 ms as FS neurons (31/204) and units with spike width >0.45 ms as putative pyramidal neurons (170/204). Units with intermediate values (0.35–0.45 ms, 3/204) were excluded from our analyses.

To quantify photoinhibition strength, we computed 'normalized spike rate' during photostimulation. For each neuron, we computed its spike rate during the photostimulus (1 s time window) and its baseline spike rate (500 ms time window before photostimulus onset). The spike rates under photostimulation were divided by the baseline spike rate. The 'normalized spike rate' thus reports the total fraction of spiking output under photostimulation. For normalized spike rate of individual neurons, each neuron's spike rate with photostimulation was normalized by dividing its baseline spike rate (*Figure 1B–D*, top). For normalized spike rate of the neuronal population (*Figure 1B–D*, bottom), the spike rates with photostimulation were first averaged across the population (without normalization) and then normalized by dividing the averaged baseline spike rate.

Bootstrap was performed over neurons to obtain standard errors of the mean. For each round of bootstrapping, repeated 1000–10000 times, we randomly sampled with replacement neurons in the dataset. We computed the means of the resampled datasets. The standard error of the mean was the standard deviation of the mean estimates from bootstrap.

## Network models

All the models we consider consist of strongly interacting leaky integrate-and-fire neurons. We first study networks of one excitatory (E) and one inhibitory (I) population. We then investigate two models comprising three inhibitory populations, namely parvalbumin positive (PV or I), somatostatin positive (SOM or S) and a third population either corresponding to the vasoactive intestinal peptide positive (VIP or V) neurons (Model 1) or to an unidentified population denoted by X (Model 2).

In all models the total number of neurons is $N = 76800$. In the two population model, 75% are excitatory and 25% inhibitory. In the four-population networks, 75% are excitatory and the number of cells is the same, $N/12$, for all GABAergic inhibitory population.

The data we seek to account for, were obtained in optogenetic experiments in which the laser diameter was substantially larger than the spatial range of neuronal interactions and comparable to the size of the cortical area were the recordings were performed. Therefore, in all models we assume for simplicity that the connectivity is unstructured: neuron $(i, \alpha)$, $(\alpha = E, I, S, V/X)$, is postsynaptically connected to neuron $(j)$ $(j, \beta)$ with probability

$$P_{ij}^{\alpha\beta} = \frac{K_{\alpha\beta}}{N_\beta} \tag{5}$$

For simplicity, we take $K_{\alpha\beta}$ the same for all populations, $K_{\alpha\beta} = K$.

*Neuron dynamics*: The dynamics between spikes of the membrane potential of the neuron $(i, \alpha)$ is given by

$$C_M \frac{dV_i^\alpha(t)}{dt} = -g_{leak}^\alpha \left( V_i^\alpha(t) - V_R \right) + I_{rec}^{\alpha i}(t) + \Lambda_{ext}^\alpha + \Lambda_{opto}^{\alpha i} \tag{6}$$

Here, $I_{rec}^{\alpha i}(t)$ is the net recurrent input into neuron $(i, \alpha)$, $\Lambda_{ext}^\alpha$ represents inputs from outside the circuit (*e.g.* thalamic excitation) to population $\alpha$, and $\Lambda_{opto}^{\alpha i}$ is the optogenetic input into neuron $(i, \alpha)$.

We assumed that the capacitance, $C_M$, is identical for all neurons and the leak conductance, $g_{leak}^\alpha$, is identical for all the cells in the same population. We take $C_M = 1\mu F.cm^{-2}$, $g_{leak}^I = 0.1mS.cm^{-2}$ and $g_{leak}^E = g_{leak}^S = g_{leak}^{V/X} = 0.05mS.cm^{-2}$.

*Equation (2)* has to be supplemented by a reset condition: if at time $t$ the membrane potential of the neuron $(i, \alpha)$ crosses the threshold $V_i^\alpha(t^-) = V_{th} = -50mV$, the neuron fires a spike and its voltage is reset to the resting potential $V_i^\alpha(t^+) = V_R = -70mV$.

*Recurrent inputs:* The net recurrent input into neuron $(i, \alpha)$ is

$$I_{rec}^{\alpha i}(t) = \sum_{\beta, j} j_{\alpha\beta} \, \epsilon_\beta \, C_{ij}^{\alpha\beta} \, S_j^{\alpha\beta}(t) \tag{7}$$

where $C^{\alpha\beta}$ is the connectivity matrix between (presynaptic) population $\beta$ and (postsynaptic) population $\alpha$, such that $C_{ij}^{\alpha\beta} = 1$ if neuron $(j, \beta)$ projects to neuron $(i, \alpha)$ and $C_{ij}^{\alpha\beta} = 0$ otherwise. The parameter $j_{\alpha\beta}$ is the strength of the interaction from neurons in population $\beta$ to neurons population $\alpha$. We assumed it to depend on the pre and postsynaptic populations only. The polarity (excitation or inhibition) of the interaction is denoted by $\epsilon_\beta$. Therefore if $\beta = E$, $\epsilon_\beta = 1$ and $\epsilon_\beta = -1$ otherwise.

The function $S_j^{\alpha\beta}(t)$ is

$$S_j^{\alpha\beta}(t) = \sum_k f_{\alpha\beta} \left( t - t_{\beta j}^k \right) \tag{8}$$

where $t_{\beta j}^k$ is the time at which neuron $(j, \beta)$ has emitted its $k^{\text{th}}$ spike, the sum is over all the spikes emitted by neuron $(j, \beta)$ prior to time $t$ and

$$f_{\alpha\beta}(t) = \frac{1}{\tau_{\alpha\beta}} e^{-t/\tau_{\alpha\beta}} \tag{9}$$

where $\tau_{\alpha\beta}$ is the synaptic time constant of the interactions between neurons in population $\beta$ and $\alpha$.

*External and optogenetic inputs*: The feedforward input, $\Lambda_{ext}^\alpha$, into the neurons in population $\alpha$ is described by inputs from $2K$ external neurons with constant firing rate $r_0 = 5$ Hz and an interaction strength $j_{\alpha 0}$, therefore, $\Lambda_{ext}^\alpha = 2K j_{\alpha 0} r_0$.

We model the ReachR photostimulation as an additional external constant input to the stimulated population. For simplicity, we assume that this input, $\Lambda_{opto}^{\alpha i} = \Lambda_{opto}^\alpha$, is the same for all stimulated neurons. Unless specified otherwise, we only consider $\Lambda_{opto}^I = \Lambda_{opto}$ and $\Lambda_{opto}^\alpha = 0$ for $\alpha \neq I$.

In qualitative agreement with *Figure 3*, and *Figures 5*, *7*, *Figure 7—figure supplement 1*, *Figure 10*; (*Hooks et al., 2015*) we take

$$\Lambda_{opto} = \Lambda_0^\alpha log\left(1 + \frac{\Gamma_{opto}}{\Gamma_0^\alpha}\right) \tag{10}$$

where $\Gamma_{opto}$ is the laser intensity and $\Lambda_0$ and $\Gamma_0$ are parameters.

## Architectures of the four-population models

The network of Model one is depicted in *Figure 4A*. In line with the results of *Pfeffer et al. (2013)*, there are no connections from PV to SOM, VIP to PC and VIP to PV neurons. There is no mutual inhibition between SOM as well as between VIP neurons. All the populations except SOM receive feedforward external input.

The interaction matrix of the network is

$$[j_{AB}\varepsilon_B] = \begin{bmatrix} j_{EE} & -j_{EI} & -j_{ES} & 0 \\ j_{IE} & -j_{II} & -j_{IS} & 0 \\ j_{SE} & 0 & 0 & -j_{SV} \\ j_{VE} & -j_{VI} & -j_{VS} & 0 \end{bmatrix} \tag{11}$$

The network of Model two is depicted in *Figure 9B*. SOM only receives projections from PCs and PV neurons. X neurons are recurrently connected and project to PCs and PV neurons. The PC and SOM populations project to the population X. All the populations except SOM receive feedforward external input.

The interaction matrix is

$$[j_{AB}\varepsilon_B] = \begin{bmatrix} j_{EE} & -j_{EI} & -j_{ES} & -j_{EX} \\ j_{IE} & -j_{II} & -j_{IS} & -j_{IX} \\ j_{SE} & -j_{SI} & 0 & 0 \\ j_{XE} & 0 & -j_{XS} & -j_{XX} \end{bmatrix} \tag{12}$$

*Numerical simulations:* The dynamics of the models was integrated numerically using a second-order Runge-Kutta scheme (*Press et al., 1986*) without spike time interpolation. Unless specified otherwise the time step was $\Delta t = 0.01$ ms and the temporally averaged firing rates were estimated over 100s.

## The balance equations

We consider recurrent networks of strongly interacting neurons (*van Vreeswijk and Sompolinsky, 1996*) in which order $\sqrt{K}$ excitatory synaptic inputs are sufficient to bring the voltage above threshold. To understand the behavior of such networks, it is imperative to analyse how it behaves when $K$ goes to infinity. To this end, we scale the interactions as

$$j_{\alpha\beta} = \frac{J_{\alpha\beta}}{\sqrt{K}} \tag{13}$$

where $J_{\alpha\beta}$ does not depend on $K$. Since a neuron receives on average $K$ inputs from each of its presynaptic populations, the total interaction from population $\beta$ to a neuron in population $\alpha$ is $J_{\alpha\beta}\sqrt{K}$. To keep the relative strength of the optogenetic input, $\Lambda_{opto}^\alpha$, as $K$ increases we take

$$\Lambda_{opto}^\alpha = I_{opto}^\alpha \sqrt{K} \tag{14}$$

where $I_{opto}^\alpha$ depends on the intensity of the laser:

$$I_{opto}^\alpha = I_0^\alpha log\left(1 + \frac{\Gamma_{opto}}{\Gamma_0^\alpha}\right) \tag{15}$$

We take: $I_0^\alpha = I_0 = 8nA$ and $\Gamma_0^\alpha = \Gamma_0 = 0.5 mW.mm^{-2}$.

The net input into the neurons must remain finite in the infinite $K$ limit. This implies that up to corrections which are of the order of $\frac{1}{\sqrt{K}}$,

$$2 J_{\alpha 0}\, r_0 + I_{opto}^{\alpha} + \sum_{\beta} J_{\alpha\beta}\, \epsilon_{\beta}\, r_{\beta} = 0 \tag{16}$$

In a *n*-population network, these *n* equations determine the *n* firing rates, $r_{\alpha}$, $\alpha \in \{1, ..., n\}$.

This set of linear equations express the fact that, for the population activities to be finite, excitatory and inhibitory inputs to the neurons must compensate. These 'balance' equations have a unique solution (unless the determinant of the matrix $J_{\alpha\beta}\epsilon_{\beta}$ is zero). To be meaningful the solution must be such that all population activities are positive. This constrains the feedforward and recurrent interaction parameters.

The stability of this balanced solution further constraints the interaction parameters and synaptic time constants. A necessary condition for the stability is that $det\left[J_{\alpha\beta}\epsilon_{\beta}\right] > 0$. This condition guarantees that the 'balanced state' is stable with respect to divergence of the firing rates. A complete study of these constraints for our LIF networks is beyond the scope of this paper.

In all the models, we study parameter ranges in which, at baseline ($I_{opto}^{\alpha} = 0$), the network operates in a stable balanced state where distributions of rates exhibit a quasi-lognormal shape and spikes are emitted irregularly as in a Poisson process (*Figure 5—figure supplement 3*; *Figure 7—figure supplement 5*; *Figure 10—figure supplement 3*). For $I_{opto}^{\alpha}$ sufficiently large, it may happen that one or more population activity reaches zero. In this case, the network evolves to a partially balanced state in which the rates of the populations that remain active satisfy a reduced set of balanced equations. For example, if we consider a solution were the rate of population $\gamma$, $r_{\gamma}$ is zero and all other rates are positive, the reduced balance equations are

$$2 J_{\alpha 0}\, r_0 + I_{opto}^{\alpha} + \sum_{\beta \neq \gamma} J_{\alpha\beta}\, \epsilon_{\beta}\, r_{\beta} = 0\, , \text{for } \alpha \neq \gamma. \tag{17}$$

Consistency of this solution leads to the requirement that the input into population $\gamma$ is hyperpolarizing.

$$2 J_{\gamma 0}\, r_0 + I_{opto}^{\gamma} + \sum_{\beta \neq \gamma} J_{\gamma\beta}\, \epsilon_{\beta}\, r_{\beta} < 0 \tag{18}$$

Note that they may be multiple self-consistent solutions which are partially balanced.

Upon photostimulation of PV, in Model 1, the balanced equations are

$$2 J_{E0}\, r_0 + J_{EE}\, r_E - J_{EI}\, r_I - J_{ES}\, r_S = 0 \tag{19.1}$$

$$2 J_{I0}\, r_0 + I_{opto}^{I} + J_{IE}\, r_E - J_{II}\, r_I - J_{IS}\, r_S = 0 \tag{19.2}$$

$$J_{SE}\, r_E - J_{SV}\, r_V = 0 \tag{19.3}$$

$$2 J_{V0}\, r_0 + J_{VE}\, r_E - J_{VI}\, r_I - J_{VS}\, r_S = 0 \tag{19.4}$$

In particular, *Equation (19.3)* implies that $r_E$ and $r_V$ are always proportional ($J_{SE}, J_{SV} > 0$).

Similarly, in Model 2, the balanced equations are

$$2 J_{E0}\, r_0 + J_{EE}\, r_E - J_{EI}\, r_I - J_{ES}\, r_S - J_{EX}\, r_X = 0 \tag{20.1}$$

$$2 J_{I0}\, r_0 + I_{opto}^{I} + J_{IE} r_E - J_{II}\, r_I - J_{IS}\, r_S - J_{IX}\, r_X = 0 \tag{20.2}$$

$$J_{SE}\, r_E - J_{SI}\, r_I = 0 \tag{20.3}$$

$$2 J_{X0}\, r_0 + J_{VE}\, r_E - J_{VS}\, r_S - J_{XX}\, r_X = 0 \tag{20.4}$$

*Equation (20.3)* implies that in this network $r_E$ and $r_I$ are always proportional ($J_{SE}, J_{SI} > 0$).

## Acknowledgements

We thank Karel Svoboda for illuminating discussions and comments on the manuscript. We are also thankful to Ran Darshan and Tohar Yarden for discussions. DH thanks Svoboda's lab. and Janelia Research Campus for their warm hospitality. This work was supported by ANR grants ANR-14-NEUC-0001–01 (CvV and DH), ANR-13-BSV4-0014-02 (DH, CvV), the ANR-09-SYSC-002–01 (DH, CvV), ANR-17-NEUC-0005 (DH, CvV, AM), the Janelia Research Campus visiting program (DH), the Helen Hay Whitney Foundation fellowship (NL), the Robert and Janice McNair Foundation (NL), Whitehall Foundation (NL), Alfred P Sloan Foundation (NL), Searle Scholars Program (NL), NIH NS104781 (NL), the Pew Charitable Trusts (NL), and Simons Collaboration on the Global Brain (#543005, NL). Work performed in the framework of the France-Israel Center for Neural Computation (CNRS/Hebrew University of Jerusalem).

## Additional information

### Funding

| Funder | Grant reference number | Author |
|---|---|---|
| Agence Nationale de la Recherche | 14-NEUC-0001-01 | Carl van Vreeswijk |
| Agence Nationale de la Recherche | 13-BSV4-0014-02 | David Hansel |
| Agence Nationale de la Recherche | 09-SYSC-002-01 | David Hansel |
| Helen Hay Whitney Foundation | | Nuo Li |
| Robert and Janice McNair Foundation | | Nuo Li |
| Alfred P. Sloan Foundation | | Nuo Li |
| National Institutes of Health | NS104781 | Nuo Li |
| Pew Charitable Trusts | | Nuo Li |
| Simons Foundation | 543005 | Nuo Li |
| Agence Nationale de la Recherche | ANR-17-NEUC-0005 | Alexandre Mahrach Carl van Vreeswijk David Hansel |

The funders had no role in study design, data collection and interpretation, or the decision to submit the work for publication.

### Author contributions

Alexandre Mahrach, Conceptualization, Software, Formal analysis, Validation, Investigation, Visualization, Methodology; Guang Chen, Conceptualization, Resources, Investigation, Methodology; Nuo Li, Conceptualization, Supervision, Funding acquisition, Investigation, Methodology; Carl van Vreeswijk, Conceptualization, Formal analysis, Supervision, Funding acquisition, Investigation, Methodology; David Hansel, Conceptualization, Formal analysis, Supervision, Funding acquisition, Investigation, Methodology, Project administration

### Author ORCIDs

Alexandre Mahrach (iD) https://orcid.org/0000-0002-9077-5808
David Hansel (iD) https://orcid.org/0000-0002-1352-6592

### Ethics

Animal experimentation: All procedures were in accordance with protocols approved by the Janelia Research Campus and Baylor College of Medicine Institutional Animal Care and Use Committee.

Decision letter and Author response
Decision letter https://doi.org/10.7554/eLife.49967.sa1
Author response https://doi.org/10.7554/eLife.49967.sa2

## Additional files

### Supplementary files
• Transparent reporting form

### Data availability

All simulation, raw, and processed data software is available on GitHub (https://github.com/Amahrach/Paper4pop; copy archived at https://github.com/elifesciences-publications/Paper4pop).

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

# Appendix 1

## Mean field theory

Let us consider a network consisting of $n$ populations (e.g. $n = 4$) receiving feedforward input, $\Lambda_{ext}^\alpha$, from an external population with constant firing rate, $r_0 r_0$, and an optogenetic input, $\Lambda_{opto}^\alpha$ (Materials and Methods). The total input into neuron $(i, \alpha)$ is

$$I_{tot}^{\alpha i}(t) = I_{rec}^{\alpha i}(t) + \Lambda_{ext}^\alpha + \Lambda_{opto}^\alpha \tag{A1}$$

If the size of the network, $N$, and mean connectivity, $K$ are large and the synaptic time constants are sufficiently small compared to the membrane time constants, one can take the diffusion approximation and neglect the temporal correlations and write

$$I_{tot}^{\alpha i}(t) = u_\alpha + \sqrt{A_\alpha}\zeta_i^\alpha + \sqrt{B_\alpha}\eta_i^\alpha(t) \tag{A2}$$

where $\zeta_i^\alpha$ is an i.i.d. Gaussian with zero mean and unit variance, and $\eta_i^\alpha(t)$ is a Gaussian white noise with zero mean and unit variance. The mean input, $u_\alpha$, is

$$u_\alpha = [<I_{tot}^{\alpha i}(t)>] = \Lambda_{ext}^\alpha + \Lambda_{opto}^\alpha + K\sum_\beta j_{\alpha\beta}\epsilon_\beta r_\beta \tag{A3}$$

where the population average firing rate of population $\beta$ is $r_\beta = [r_j^\beta]$ and $r_j^\beta$ is the firing rate of the neuron $(j, \beta)$. Here $<.>$ denotes temporal average (i.e. over $\eta_i^\alpha(t)$) and $[.]$ is the average over the quenched disorder $(\zeta_i^\alpha)$. The latter stems from heterogeneities in the in-degree of the inputs into the neurons.

In *Equation (A2)*, $A_\alpha$ is the variance of the quenched disorder which is given by

$$A_\alpha = [<I_{tot}^{\alpha i}(t)>^2 - u_\alpha^2] = K\sum_\beta j_{\alpha\beta}^2 q_\beta \tag{A4}$$

while $B_\alpha$ is the variance of the temporal fluctuations (*Van Vreeswijk and Sompolinsky, 2005*; *Roxin et al., 2011*)

$$B_\alpha = \frac{1}{\tau_m^\alpha}\lim_{\Delta t\to 0}\left[\frac{1}{\Delta t}\int_t^{t+\Delta t}\left(dt' I_{tot}^{\alpha i}(t') - <I_{tot}^{\alpha i}(t')>\right)^2\right] \tag{A5}$$

In *Equation (A4)*, $q_\beta = [(r_j^\beta)^2]$.

*Equations (A4-A5)* have to be supplemented with the expression of the input-output transfer function which relates the average firing rate, $r_i^\alpha$, to the statistics of $I_{tot}^{\alpha i}(t)$,

$$r_i^\alpha = \Phi_\alpha(u_\alpha + \sqrt{A_\alpha}\,\zeta_i^\alpha, B_\alpha) \tag{A6}$$

$$r_\alpha = \int D\zeta\,\Phi_\alpha(u_\alpha + \sqrt{A_\alpha}\zeta, B_\alpha) \tag{A7}$$

$$q_\alpha = \int D\zeta\,\Phi_\alpha(u_\alpha + \sqrt{A_\alpha}\zeta, B_\alpha)^2 \tag{A8}$$

where $D\zeta = \frac{1}{\sqrt{2\pi}}e^{-\zeta^2/2}$, and $\Phi_\alpha$ is given by *Capocelli and Ricciardi (1971)*

$$\Phi_\alpha(x,y) = \left\{\sqrt{\frac{\pi\tau_m^\alpha}{y}}\int_{X_\alpha^+}^{X_\alpha^-}dw\,e^{w^2}erfc(w)\right\}^{-1} \tag{A9}$$

where $X_\alpha^- = \frac{x - g_{leak}^\alpha V_R}{\sqrt{y}}$, $X_\alpha^+ = \frac{x - g_{leak}^\alpha V_{Th}}{\sqrt{y}}$ and $\tau_\alpha = \frac{C_M}{g_{leak}^\alpha}$ is the membrane time constant of the neurons in population $\alpha$.

With $j_{\alpha\beta} = \frac{J_{\alpha\beta}}{\sqrt{K}}$, $\Lambda_{ext}^\alpha = 2\sqrt{K}$ and $\Lambda_{opto}^\alpha = I_{opto}^\alpha\sqrt{K}$ (see Materials and methods), we obtain

$$u_\alpha = \sqrt{K}\left(2J_{\alpha 0}\, r_0 + I^\alpha_{opto} + \sum_\beta J_{\alpha\beta}\, \epsilon_\beta\, r_\beta\right) \tag{A10}$$

$$A_\alpha = \sum_\beta J^2_{\alpha\beta} q_\beta \tag{A11}$$

$$B_\alpha = \frac{1}{\tau^\alpha_m}\sum_\beta J^2_{\alpha\beta} r_\beta \tag{A12}$$

For finite, but large K, the average activity of population $\alpha$ is

$$r_\alpha = \Psi_\alpha[u_\alpha, A_\alpha, B_\alpha] \tag{A13}$$

where $\Psi_\alpha$ is the right hand-side of **Equation (A7)**.

In the limit where $u_\alpha \to -\infty$, it can be shown that

$$\Psi_\alpha[u_\alpha, A_\alpha, B_\alpha] \sim -\frac{u_\alpha}{\tau^\alpha_m\sqrt{\pi}}\frac{B_\alpha}{(2A_\alpha + B_\alpha)^{3/2}}e^{-\frac{u^2_\alpha}{2A_\alpha + B_\alpha}} \tag{A14}$$

In the large K limit, the activities, $r_a$, have to satisfy a set of n linear balance equations (**Equation (12)**, Materials and methods) and are given by

$$r_\alpha = -\epsilon_\alpha \sum_\beta [J^{-1}]_{\alpha\beta}\left(2\, J_{\beta 0}\, r_0 + I^\beta_{opto}\right) \tag{A15}$$

We define the susceptibility matrix, $X_{\alpha\beta}$, as the derivative of the activity, $r_a$, with respect to $I^\beta_{opto}$,

$$\chi_{\alpha\beta} = -\epsilon_\alpha [J^{-1}]_{\alpha\beta} \tag{A16}$$

At baseline $\left(I^\beta_{opto} = 0\right)$, the positivity of $r_\alpha, \forall\alpha$ imposes conditions on the recurrent and feedforward interaction strengths, $J_{\alpha\beta}$ and $J_{\alpha 0}$. The requirement that there are no 'partially' balanced solutions for which one or more of the n populations is inactive or saturates and the stability of the balanced solution imposes further constraints.

## Two-population model
### Large K limit

For a two-population (one excitatory E and one inhibitory I) network, solving **Equation (13)** gives for a perturbation, $I_{opto}$, upon I,

$$r_E = \frac{2(J_{II}J_{E0} - J_{EI}J_{I0})r_0 - J_{EI}I_{opto}}{\Delta} \tag{A17}$$

$$r_I = \frac{2(J_{IE}J_{E0} - J_{EE}J_{I0})r_0 - J_{EE}I_{opto}}{\Delta} \tag{A18}$$

where $\Delta = J_{EI}J_{IE} - J_{EE}J_{II}$.

The requirement that at baseline the network state is fully balanced and stable implies that

$$\frac{J_{E0}}{J_{I0}} > \frac{J_{EI}}{J_{II}} > \frac{J_{EE}}{J_{IE}} \tag{A19}$$

Therefore, $\Delta > 0$.

The susceptibilities with respect to a perturbation of I are

$$\chi_{EI} = \frac{-J_{EI}}{\Delta} \tag{A20}$$

$$\chi_{II} = \frac{-J_{EE}}{\Delta} \qquad (A21)$$

which both are negative. Therefore, $r_E$ and $r_I$ *decrease* linearly with $I_{opto}$, that is the response of the I population is paradoxical.

It is useful to consider the susceptibilities normalized to baseline rate

$$\bar{\chi}_{EI} = -\frac{J_{EI}}{2(J_{II}J_{E0} - J_{EI}J_{I0})r_0} \qquad (A22)$$

$$\bar{\chi}_{II} = -\frac{J_{EE}}{2(J_{IE}J_{E0} - J_{EE}J_{I0})r_0} \qquad (A23)$$

*Equation (A19)* implies that, $|\bar{\chi}_{EI}|$ is larger than $|\bar{\chi}_{II}|$.

Moreover, whereas $\bar{\chi}_{EI}$ is independent of $J_{EE}$, $\bar{\chi}_{II}$ depends on $J_{EE}$. When $J_{EE} = 0$, $\bar{\chi}_{II}$ is zero: the PV activity is insensitive to $I_{opto}$.

The identity of the two normalized susceptibilities can only be achieved with a fine-tuning of the interaction parameters such that $\Delta \simeq 0$ for

$$J_{EE} \simeq J_{EI}J_{IE}/J_{II} \qquad (A24)$$

Concurrently, as $J_{EE} \to J_{EI}J_{IE}/J_{II}$, the activity of the two populations diverge as $\frac{1}{\Delta}$ with a constant ratio equal to $\frac{J_{IE}}{J_{II}}$. Thus, to keep the activities finite, $2(J_{II}J_{E0} - J_{EI}J_{I0})r_0$ and $2(J_{IE}J_{E0} - J_{EE}J_{I0})r_0$ must also tend to zero.

Finally, if $I_{opto} = I^*_{opto} \equiv 2(J_{E0}J_{II}/J_{EI} - J_{I0})r_0$, $r_E$ vanishes (*Figure 3—figure supplement 1*). When $I_{opto} > I^*_{opto}$, the balance between the total external excitatory (optogenetic+feedforward) and recurrent inhibitory inputs into I implies that $r_I$ linearly increases with $I_{opto}$ and the slope is $1/J_{II}$.

## Finite K corrections to $r_E$ and $r_I$ near $I^*_{opto}$

When $K$ is finite, $r_I$ starts to increase with $I_{opto}$ when $r_E$ is exponentially small in $K$. To show that, we have to derive the leading order correction to the activities near $I^*_{opto}$.

We make the ansatz that when $I_{opto} = I^*_{opto} + \delta I \sqrt{\frac{log(K)}{K}}$, $r_E = \nu_E \frac{\sqrt{log(K)}}{K}$ and $r_I = r^\infty_I + \nu_I \sqrt{\frac{log(K)}{K}}$, where $\nu_E$ and $\nu_I$ are $O(1)$ and $r^\infty_I = 2J_{E0}r_0/J_{EI}$ is the inhibitory firing rate at $I_{opto} = I^*_{opto}$ in the large $K$ limit.

To leading order:

$$r^\infty_I = \Psi_E\left[\sqrt{log(K)}(\delta I + J_{IE}\nu_E - J_{II}\nu_I), A^\infty_I, B^\infty_I\right] \qquad (A25.1)$$

$$\nu_E\sqrt{\frac{log(K)}{K}} = \Psi_E\left[\sqrt{log(K)}(J_{EE}\nu_E - J_{EI}\nu_I), A^\infty_E, B^\infty_I\right] \qquad (A25.2)$$

where $A^\infty_\alpha$ and $B^\infty_\alpha$, $\alpha \in \{E, I\}$, are the variance of the temporal and quenched noise in the large $K$ limit (*Equations (A11-A12)*).

*Equation (A25.1)* implies that

$$\delta I + J_{IE}\nu_E - J_{II}\nu_I = O\left(\frac{1}{\sqrt{log(K)}}\right) \qquad (A26)$$

Together with *Equation (A25.2)* one obtains

$$\nu_E\sqrt{\frac{log(K)}{K}} = \Psi_E\left[\left(-(J_{EI}\delta I + \nu_E\Delta)\sqrt{log(K)}/J_{II}\right), A^\infty_E, B^\infty_I\right] \qquad (A27)$$

where $\Delta = J_{EI}J_{IE} - J_{EE}J_{II}$.

For large $K$,

$$\frac{\nu_E}{\sqrt{K}} = \frac{Q}{J_{II}}(J_{EI}\delta I + \nu_E\Delta)e^{-\frac{(J_{EI}\delta I + \nu_E\Delta)^2 log(K)}{(2A_E^\infty + B_E^\infty)J_{II}^2}} \tag{A28}$$

where $Q = \frac{1}{\tau_m^E\sqrt{\pi}}\frac{B_E^\infty}{\left(2A_E^\infty + B_E^\infty\right)^{3/2}}$.

Since $\nu_E$ must be positive, $(J_{EI}\delta I + \nu_E\Delta)$ must also be positive, *Equation (A28)* then implies that to leading order

$$\nu_E = \frac{1}{\Delta}\left(J_{II}\sqrt{A_E^\infty + \frac{B_E^\infty}{2}} - J_{EI}\delta I\right) \tag{A29}$$

Hence, $\nu_I$ is

$$\nu_I = \frac{1}{\Delta}\left(J_{IE}\sqrt{A_E^\infty + \frac{B_E^\infty}{2}} - J_{EE}\delta I\right) \tag{A30}$$

Therefore, both $\nu_E$ and $\nu_I$ decrease with $\delta I$. This holds for $\delta I \le \frac{J_{II}}{J_{EI}}\sqrt{A_E^\infty + \frac{B_E^\infty}{2}}$. Beyond this range $r_E$ is exponentially small, $\nu_I = \frac{\delta I}{J_{II}}$ and $r_I$ increases with $I_{opto}$.

In conclusion, when the response of the I population is minimum the firing rate of the excitatory population is exponentially small in $K$.

## Four-population model: Model 1
### Large $K$ limit

In Model 1, the population susceptibilities in response to a perturbation of the PV population are given by *Equation (A16)*

$$\chi_{EI} = J_{SV}(J_{EI}J_{VS} - J_{ES}J_{VI})/\Delta \tag{A31}$$

$$\chi_{II} = J_{SV}(J_{EE}J_{VS} - J_{ES}J_{VE})/\Delta \tag{A32}$$

$$\chi_{SI} = J_{SV}(J_{EI}J_{VE} - J_{EE}J_{VI})/\Delta \tag{A33}$$

$$\chi_{VI} = \frac{J_{SE}}{J_{SV}}\chi_{EI} \tag{A34}$$

where $\Delta = det([J_{AB}\epsilon_B])$.

Note, in this model we do not take into account any PV to SOM connections. Nevertheless even If one includes these, the expressions of the PC and PV susceptibility will only differ by a scaling factor from the ones in A31 and A32 (because of $\Delta$) and therefore their sign will depends on the same conditions than A31 and A32.

Interestingly, for stable solutions ($\Delta > 0$), then $\chi_{II} > 0$ implies that $J_{EE} J_{VS} > J_{ES} J_{VE}$. while $\chi_{EI} < 0$ implies that $J_{ES} J_{VI} > J_{EI} J_{VS}$. Therefore, $J_{EE} J_{VS} J_{VI} > J_{VE} J_{ES} J_{VI}$. and $J_{ES} J_{VI} J_{VE} > J_{EI} J_{VS} J_{VE}$. Combining the latter one has $J_{EE} J_{VS} J_{VI} > J_{EI} J_{VS} J_{VE}$. Therefore, $J_{EE} J_{VI} > J_{EI} J_{VE}$ which is equivalent to $\chi_{SI} < 0$.

Similarly one can show that if $\chi_{EE} > 0$ and $\chi_{IE} < 0$ necessarily $\chi_{SE} > 0$. Let us consider a particular set of parameters for which a stable balanced solution exists when $J_{EE} = 0(\Delta(0) > 0)$.

The susceptibility $\chi_{II}$ as a function of $J_{EE}$ is

$$\chi_{II}(J_{EE}) = J_{SV}\frac{J_{VS}J_{EE} - J_{VE}J_{ES}}{\Delta(J_{EE})} \tag{A35}$$

$$\Delta(J_{EE}) = -\hat{\chi}_{EE}J_{EE} + \Delta(0) \tag{A36}$$

where $\hat{\chi}_{EE} \equiv \chi_{EE}.\Delta(J_{EE}) = J_{SV}(J_{VI}J_{IS} - J_{II}J_{VS})$, is the numerator in the susceptibility $\chi_{EE}$.

In our models, we assumed $\chi_{EE} > 0$. When $J_{EE} = 0$, $\Delta(0)$ is positive thus, $\chi_{II}(0) < 0$. As $J_{EE}$ increases, the sign of $\chi_{II}(J_{EE})$ depends on the order relationship between two quantities. The first one, $J_{EE}^*$, is the value of $J_{EE}$ for which the numerator in **Equation (A35)** changes sign

$$J_{EE}^* = \frac{J_{VE} J_{ES}}{J_{VS}} \tag{A37}$$

The second one, $J_{EE}^c$, is defined by $\Delta(J_{EE}^c) = 0$

$$J_{EE}^c = \frac{\Delta(0)}{\chi_{EE}} \tag{A38}$$

Therefore, for $J_{EE} > J_{EE}^c$, the dynamics is unstable. Two cases can be distinguished:

1. If $J_{EE}^* < J_{EE}^c$, then $\chi_{II}$ is an increasing function of $J_{EE}$. It is negative if $J_{EE} < J_{EE}^*$ and becomes positive for $J_{EE} > J_{EE}^*$.
2. If $J_{EE}^* < J_{EE}^c$, $\chi_{II}$ is a decreasing function of $J_{EE}$ and is negative in all the region where the dynamics is stable.

The derivative of $\chi_{II}$, (**Equation (A35)**), with respect to $J_{EE}$, has the same sign as $\chi_{EI}\chi_{IE}$. Therefore, $\chi_{EI}\chi_{IE}$ is positive in the first case and negative in the second.

Experimental data shows that the activity of the PC population decreases upon PV photostimulation, *i.e.*, $\chi_{EI} < 0$. Therefore, if $\chi_{II} > 0$ as in ALM layer 2/3, $\chi_{IE}$ must be negative, *i.e.*, the activity of the PV population decreases upon PC photostimulation.

## Finite $K$

When $I_{opto}$ is sufficiently strong, a fully balanced solution $(r_\alpha > 0, \forall \alpha)$ no longer exists in our case $r_E = r_V = 0$ for $I_{opto} > I_{opto}^*$ where

$$I_{opto}^* = \frac{J_{E0}(J_{IS}J_{VI} - J_{II}J_{VS}) + J_{I0}(J_{EI}J_{VS} - J_{ES}J_{VI}) + J_{V0}(J_{ES}J_{II} - J_{EI}J_{IS})}{(J_{ES}J_{VI} - J_{EI}J_{VS})}$$

To understand the network behavior after this point we need to consider finite $K$ corrections.

Since the PC and VIP population activities decrease with $I_{opto}$, when $I_{opto}$ is sufficiently large and due to the balance of the SOM input, $r_E$ and $r_V$ will both be at most $O\left(\frac{1}{\sqrt{K}}\right)$. Let us write: $r_E \equiv \frac{\nu_E}{\sqrt{K}}$ and $r_V \equiv \frac{\nu_V}{\sqrt{K}}$ where $\nu_E$ and $\nu_V$ are at most $O(1)$.

One should consider four cases:

1) $\nu_E$ and $\nu_V$ are $O(1)$

In this case, the average net input into the SOM population, $u_S = J_{SE}\nu_E - J_{SV}\nu_V$, is $O(1)$ and the temporal fluctuations, $B_S$, and heterogeneities, $A_S$, are negligible. If $u_S$ is larger than the rheobase, $(V_{th} - V_R)/g_{leak}^S$, $r_S$ is also $O(1)$. Otherwise, $r_S = 0$.

Because $\nu_E$ and $\nu_V$ are $O(1)$, $u_E$ and $u_V$ are $o\left(\frac{1}{\sqrt{K}}\right)$. Thus, to leading order,

$$2J_{E0}r_0 - J_{EI}r_I - J_{ES}r_S = 0 \tag{A39}$$

$$2J_{V0}r_0 - J_{VI}r_I - J_{VS}r_S = 0 \tag{A40}$$

Moreover, the balance of the PV population implies that

$$2J_{I0}r_0 + I_{opto} - J_{II}r_I - J_{IS}r_S = 0 \tag{A41}$$

Thus, there are three linear equations (**Equations (A39-A41)**) for two unknowns ($r_I$ and $r_s$). These cannot be satisfied and hence, in this case, there is no consistent solution.

2) $\nu_E = o(1)$ and $\nu_V = O(1)$

Here, to leading order, $u_S = -J_{SV}\nu_V < 0$, while $A_S = B_S = 0$. As a result, to leading order, $r_S = 0$. The activity of the PV population is then

$$r_I = \left(2J_{I0}r_0 + I_{opto}\right)/J_{II} \tag{A42}$$

Because $\nu_V$ is $O(1)$,

$$2J_{V0}r_0 - J_{VI}r_I = 0 \tag{A43}$$

*Equations (A42, A43)* cannot both be satisfied. This solution is also inconsistent.

3) $\nu_E = O(1)$ and $\nu_V = o(1)$

In this case $u_S = J_{SE}\nu_E > 0$ and therefore $r_S$ can be $O(1)$. *Equations (A39) and (A41)* imply

$$2J_{E0}r_0 - J_{EI}r_I - J_{ES}r_S = 0 \tag{A44}$$

$$2J_{I0}r_0 + I_{opto} - J_{II}r_I - J_{IS}r_S = 0 \tag{A45}$$

which determine $r_I$ and $r_S$ as $r_I = \frac{(J_{ES}J_{I0} - J_{IS}J_{E0})r_0 + J_{ES}I_{opto}}{J_{ES}J_{II} - J_{EI}J_{IS}}$ and $r_S = \frac{(J_{II}J_{E0} - J_{EI}J_{E0})r_0 - J_{EI}I_{opto}}{J_{ES}J_{II} - J_{EI}J_{IS}}$.

Provided that the parameters are such that they are positive, $\nu_E$ is given by

$$r_S = \Psi_S[J_{SE}\nu_E, 0, 0] \tag{A46}$$

Finally, since $\nu_V = o(1)$ consistency implies that

$$2J_{V0}r_0 - J_{VI}r_I - J_{VS}r_S < 0 \tag{A47}$$

This solution is valid for a finite range of $I_{opto}$. It exists as long as $r_s > 0$ which implies that $J_{E0}\frac{J_{II}}{J_{EI}} - J_{I0} > I_{opto} > I^*_{opto}$.

4) $\nu_E = o(1)$ and $\nu_V = o(1)$

Here, $u_S = A_S = B_S = 0$ and thus, $r_S = 0$. This solution exists only for sufficiently large $I_{opto}$ such that $u_E$ and $u_V$ are $O(\sqrt{K})$ and negative. Therefore, PV is the only active population and $r_I$ is given by *Equations (A40)*.

In conclusion, in this model at the minimum of $r_I$, $r_E$ is of order $\frac{1}{\sqrt{K}}$ in contrast to the two-population case where $r_E$ is exponentially small in $K$.

## Four-population model: Model 2
### Large $K$ limit

To get insights on the network architecture that could explain the proportional paradoxical effect observed in layer 5 of ALM and S1, we first considered a three-population network consisting of the PC, PV and SOM populations (*Figure 9A*).

In this network, the population activities are

$$r_E = J_{SI}\frac{2(J_{ES}J_{I0} - J_{IS}J_{E0})r_0 + J_{ES}I_{opto}}{\Delta} \tag{A48}$$

$$r_I = \frac{J_{SE}}{J_{SI}}r_E \tag{A49}$$

$$r_S = \frac{2((J_{II}J_{SE} - J_{IE}J_{SI})J_{E0} - (J_{EI}J_{SE} - J_{EE}J_{SI})J_{I0})r_0 - (J_{EI}J_{SE} - J_{EE}J_{SI})I_{opto}}{\Delta} \tag{A50}$$

where $\Delta = (J_{II}J_{SE} - J_{IE}J_{SI})J_{ES} + (J_{EE}J_{SI} - J_{EI}J_{SE})J_{IS} > 0$.

The full balance of the network activities implies

$$\frac{J_{ES}}{J_{IS}} > \frac{2J_{E0}r_0}{2J_{I0}r_0 + I_{opto}} > \frac{J_{EI}}{J_{II}} \tag{A51}$$

The inequality on the left side stems from the positivity of the rates. The inequality on the right side stems from the fact that the balanced state is the only solution of the dynamics, namely that no

partially balanced solution (in particular, $r_E = 0$, $r_I = O(1)$ and $r_S = 0$ and $r_E = 0$, $r_I = O(1)$ and $r_S = O(1)$) exists.

$r_E$ and $r_I$ are proportional **Equations (A49)** and increase with $I_{opto}$. As a consequence, the network never exhibits the paradoxical effect.

In this three-population network, the proportionality of $r_E$ and $r_I$ stems from the balance of inputs into the SOM population. To account for the proportional *paradoxical effect*, we consider a network model with an additional inhibitory population, denoted X (**Figure 9B**). Because in this network the SOM neurons only receive inputs from PCs and PV neurons, here, the balance of the SOM input also ensure the proportionality of $r_E$ and $r_I$.

The susceptibilities upon PV stimulation are

$$\chi_{EI} = J_{SI}(J_{ES}J_{XX} - J_{EX}J_{XS})/\Delta \tag{A52}$$

$$\chi_{II} = \frac{J_{SE}}{J_{SI}}\chi_{EI} \tag{A53}$$

$$\chi_{SI} = (J_{EE}J_{SI}J_{XX} - J_{XE}J_{SI}J_{XE} - J_{EI}J_{SE}J_{XX})/\Delta \tag{A54}$$

$$\chi_{XI} = (J_{ES}J_{SI}J_{XE} + J_{EI}J_{SE}J_{XS} - J_{EE}J_{SI}J_{XS})/\Delta \tag{A55}$$

where $\Delta = det([J_{AB}\epsilon_B])$ (see Material and methods).

Paradoxicality implies that

$$J_{EX} > J_{EX}^* \equiv \frac{J_{ES}J_{XX}}{J_{XS}} \tag{A56}$$

The susceptibilities upon PC stimulation are

$$\chi_{EE} = J_{SI}(J_{IX}J_{XS} - J_{IS}J_{XX})/\Delta \tag{A57}$$

$$\chi_{IE} = \frac{J_{SE}}{J_{SI}}\chi_{EE} \tag{A58}$$

$$\chi_{SE} = (J_{IX}J_{SI}J_{XE} + J_{II}J_{SE}J_{XX} - J_{IE}J_{SI}J_{XX})/\Delta \tag{A59}$$

$$\chi_{XE} = (J_{IE}J_{SI}J_{XS} - J_{IS}J_{SI}J_{XE} - J_{II}J_{SE}J_{XS})/\Delta \tag{A60}$$

Therefore, the PC population activity increases upon PC stimulation if

$$J_{IX}J_{XS} > J_{IS}J_{XX} \tag{A61}$$

One can find a range of parameters (*e.g.* **Figure 9C**) such that:

1. The relative decrease in the SOM population is larger than that in the E and I populations. As a consequence, as $I_{opto}$ is increased, $r_S$ approaches zero when the PC and PV activities are still finite.
2. As $I_{opto}$ is increased further, the network settles into a partially balanced state where $r_E$, $r_I$ and $r_X$ are finite and $r_I$ increases with $I_{opto}$, while $r_E$ continues to decrease.

Thus, $r_I$ reaches its minimum value when $r_E$ is finite even in the large $K$ limit.

