## [Decision Letter]

**Acceptance summary:**

Hansel and colleagues investigate the role of feedback in the stabilization of neuronal activity in cortex. The simplest models for feedback involve two populations of neurons, a single population of inhibitory neurons and a population of excitatory neurons. This class of models is sufficient to explain seeming paradoxical effects within the realm of cortical circuits, such as decreased overall inhibitory cell activity upon excitatory perturbation of inhibitory neurons. However, the authors show that "two population" models fail to offer robust solutions for the responses they observe in new, optogenetic perturbation experiments on neuronal dynamics in mouse sensory and motor cortices. Rather, a more complex model, with feedback among three classes of inhibitory neurons and associated constrained connectivity, along with a population of excitatory neurons, is needed. The "four population" models give rise to a second-order feature, disynaptic inhibition, to achieve stabilization of neuronal activity.

**Decision letter after peer review:**

Thank you for submitting your article "Mechanisms underlying the response of mouse cortical networks to optogenetic manipulation" for consideration by *eLife*. Your article has been reviewed by three peer reviewers, and the evaluation has been overseen by a Reviewing Editor and Michael Frank as the Senior Editor. The following individuals involved in review of your submission have agreed to reveal their identity: David Golomb (Reviewer #1); Misha Tsodyks (Reviewer #3).

The reviewers have discussed the reviews with one another and the Senior Editor has drafted this decision to help you prepare a revised submission.

Summary:

The "paradoxical effect" is the phenomenon that stimulation of an inhibitory neuronal population decreases the average firing activity of neurons in that population. The conditions for its existence have been under debate, and the mechanistic impacts of various types of inhibitory interneurons has not been elucidated. The manuscript of Mahrach et al. addresses these two important issues, and is a big step forward in understanding cortical dynamics. The authors present new experimental data collected from PC and PV neurons in the anterior lateral motor cortex (ALM; layer 2/3 and layer 5) and the barrel cortex (S1; layer 5) during photostimulation of PV neurons. The data show paradoxical effects in layer 5 (i.e., at a low light intensity, increasing the stimulus decreases PV firing rates proportionally to that of PC cells), while showing non-paradoxical effects in layer 2/3. The results are novel and contradict the widespread notion that the paradoxical effect is an evidence for stabilization by inhibition, and the modeling suggests an architecture consistent with the results.

Essential revisions:

1) The manuscript is based on the analytical calculations, and it is expected that readers will try to replicate them. Therefore, it is important that their description will be as clear and as detailed as possible, which will enhance the readability of the paper.

2) Overall the paper rests on comparing how photostimulation of PV neurons drives population-wide rate activity between experiments and several different models. The paper would be strengthened if the authors used statistical tests to show that layer 2/3 is significantly different than layer 5, as well as showing that the proportional decrease of PV and PC cells is a robust observation. We would like to see some actual statistical comparisons between the data and the models. More to the point, the distribution of firing rates for baseline vs. photostimulation in Figure 2 (experiment) should be compared in some statistical way to those in Figure 6 (Model 1, JEE>JEE*), Figure 8 (Model 1, JEE<JEE^*^), and Figure 11 (Model 2). The pie charts in Figures 6, 8, 11 are nice but they should have some confidence bounds and then compared to the equivalent pie charts of Figure 2. This will go a long way in helping the narrative of the paper where models are accepted or rejected based on the three datasets shown in Figures 1-2.

3) The data are used to assess, dismiss, and propose possible network architectures. Using analytical results derived from a balanced network framework and simulations, they settle on significantly different architectures for each layer. However, some of the reviewers were not convinced that the more complex networks capture enough of the properties present in the data, and were not persuaded by their argument that Model 1 should be dismissed with regards to layer 5, and wary of the presence of inhibitory population 'X' in Model 2.

The dismissal of Model 1 with regards to layer 5 needs additional details. Currently, the authors dismiss Model 1 since it cannot robustly capture the fact that PC and PV activity decreases proportionally in the paradoxical regime.

a) Attaching some quantitative measures to the phrase "decreasing proportionally" would assist with this argument. Figure 7A and Figure 1B lower right potentially look similar "enough".

b) Further, Figure 7—figure supplement 3 shows that the right parameters yield a great proportional decrease. I believe that this parameter regime is potentially small, but ideally it would be nice to include a figure showing how small (to my knowledge, one like Figure 3—figure supplement 3 doesn't exist for this parameter regime).

4) Relatedly, the authors propose a series of improved network architectures over the traditional E,I-two population model that incorporate known interneuron subclasses (PV, SOM, and VIP). However, with every improvement the more complex models bring, I find additional questions regarding the data that is not captured.

a) For example, Model 1 provides a network that is able to provide a non-paradoxical response such that PV neurons increase in rate at low light stimulations. This is not achievable by the E,I network. However, the heterogeneity of neurons seen as a function of stimulus strength in Model 1 seems different than experimental results (relevant figures for comparison: Figure 1B, Figure 3C, and Figure 5A), and is not discussed in the text. I would've hoped that adding such a large change in the network would've been able to better capture the data.

b) I would also like to directly compare Figure 2 (left column) with Figure 6 (left or right column). The distribution of variability of firing rates seems different between Model 1 and the experimental results. However, the strength of stimulus is different in Figure 2 than in Figure 6 (0.5 for the experiments than 0.3 and 0.9 for the simulations), so I would request that Figure 2 be remade with one of these stimulation strengths. Having a similar pie chart appear in Figure 2 would also be helpful. Lastly, marginal histograms in both figures would assist with comparison.

Also, in regards to Figure 6, the authors comment that "Remarkably, even for 0.9 mW/mm^2^, some of the PCs show an activity increase." However, this was not observed in experiments.

5) To capture the final missing piece of the data (i.e., the proportional decrease of PC and PV cells), the authors propose an entirely new network architecture with inhibitory interneuron 'X'.

a) In addition to this new type of neuron, the authors must also add a connection from PV to SOM cells (otherwise, r_E_ = 0 in the large network limit). While the authors suggest that 'X' may be chandelier cells, they do not discuss why this added connection from PV to SOM would be present in layer 5 but not layer 2/3.

b) Similar to my above comment, I would expect that such a drastic change in the network would be able to capture additional features of the experimental data, but the heterogeneity present in Figure 10A is drastically different than Figure 1B, right column.

Simulations must be made publicly available.

---

## [Author Response]

Essential revisions:1) The manuscript is based on the analytical calculations, and it is expected that readers will try to replicate them. Therefore, it is important that their description will be as clear and as detailed as possible, which will enhance the readability of the paper.

We have improved the clarity of the analytical calculations.

2) Overall the paper rests on comparing how photostimulation of PV neurons drives population-wide rate activity between experiments and several different models.The paper would be strengthened if the authors used statistical tests to show that layer 2/3 is significantly different than layer 5, as well as showing that the proportional decrease of PV and PC cells is a robust observation.

We have included a statistical comparison between ALM layer 2/3 and layer 5 and S1 in the revised Figure 1. At moderate light intensity (0.5 mW/mm^2^), the relative spike rate of PV neurons are significantly different (p<0.005, unpaired t-test, Figure 1E).

As mentioned in the preamble, statistical analysis shows that the ratio of the slopes of the normalized response of the PC and PV populations is 1 ± 0.29 in S1 while in ALM layer 5 it is 0.62 ± 0.28. Therefore, ALM layer 5 *can* be described by Model 1.

We have modified the Discussion in accordance.

We would like to see some actual statistical comparisons between the data and the models. More to the point, the distribution of firing rates for baseline vs. photostimulation in Figure 2 (experiment) should be compared in some statistical way to those in Figure 6 (Model 1, JEE>JEE*), Figure 8 (Model 1, JEE<JEE*), and Figure 11 (Model 2).

The main purpose of our work is to gain an understanding of the network properties that give rise to the response to optical stimulation of the PV neurons. We investigate which network architecture may account in a qualitative manner for the responses observed experimentally. While our network models give rise to substantial heterogeneity, we did not try to match this heterogeneity with experimental data.

First of all, the number of PV neurons we recorded from in our experiment is too small to reliably estimate the heterogeneity in their responses. Secondly, to match this heterogeneity we would have needed quantitative data on the input-output relations of the PC and PV populations. Thirdly, even if this data were available, we would have to use more complicated neuronal models which would have made the mathematical analysis prohibitively complicated, limiting the investigation to simulations only and thus obscuring the mechanisms. Our goal instead was to investigate these mechanisms. As a result, a statistical comparison between model and experimental heterogeneity is not appropriate. We have added these considerations to the revised manuscript in the beginning of the “Limitations” subsection. We write:

“We give here a qualitative account for the mechanisms underlying the responses of different cortical areas to optical stimulation. […] Moreover, it would necessitate the use of more complicated neuronal models making the mathematical analysis intractable, limiting the investigation to simulations only and thus obscuring the mechanisms.”

The pie charts in Figures 6, 8, 11 are nice but they should have some confidence bounds and then compared to the equivalent pie charts of Figure 2.

The pie charts for the modelling are based on a very large number of neurons in the model networks (57600 for PCs and 6400 for PV neurons). As a result, in the analysis of the simulations, the error bars are tiny. We have included equivalent pie charts in the revised Figure 2.

This will go a long way in helping the narrative of the paper where models are accepted or rejected based on the three datasets shown in Figures 1-2.

We showed that a two-population model cannot account for the response in either ALM layer 2/3 and layer 5 as well as in S1. In contrast, we showed that the average response of ALM layer 2/3 and layer 5 can be accounted for by a four-population network with connectivity similar to that reported for V1 (our Model 1). If we fine-tuned the parameters, this model can also account for the proportional decrease in PC and PV neurons average activities observed in S1. For S1 we proposed an alternative network architecture (our Model 2) in which this proportional decrease occurs robustly without any fine-tuning. Nevertheless, we do not reject Model 1 in the manuscript. We have clarified this point in the Discussion. We now write for Model 1:

“In Model 1, whether the network exhibits a paradoxical effect depends on the value of the ratio ρ=JEE/JEE*where JEE*≡JVEJES/JVS. Here, Jαβ,α,β∈{E,S,V}, is the strength of the connection from population β to population α. When *ρ* > 1, the PV response is non-paradoxical and its activity increase can be substantial well before suppression of the PC activity. On the other hand when *ρ* < 1, the PV response is paradoxical and the PV activity reaches its minimum for light intensities at which the PCs are still substantially active. […] The interactions *J_VE_,J_ES_
* and *J_VS_
* are likely to be layer dependent (Jiang et al., 2015)  and therefore so is the value of JEE^*^”

For Model 2 we write:

“Similar to ALM layer 5, the PV response in S1 is paradoxical. Remarkably however, in S1 the relative suppression of the PC and PV activities is the same for low light intensity. Model 1 can account for this feature only when the interaction parameters are fine tuned. […] Furthermore, it can equally well account for the fact that in S1 the PV activity reaches its minimum when the PC population is active.”

3) The data are used to assess, dismiss, and propose possible network architectures. Using analytical results derived from a balanced network framework and simulations, they settle on significantly different architectures for each layer. However, some of the reviewers were not convinced that the more complex networks capture enough of the properties present in the data, and were not persuaded by their argument that Model 1 should be dismissed with regards to layer 5, and wary of the presence of inhibitory population 'X' in Model 2.The dismissal of Model 1 with regards to layer 5 needs additional details. Currently, the authors dismiss Model 1 since it cannot robustly capture the fact that PC and PV activity decreases proportionally in the paradoxical regime.a) Attaching some quantitative measures to the phrase "decreasing proportionally" would assist with this argument. Figure 7A and Figure 1B lower right potentially look similar "enough".

We compared the change in relative spike rate with light intensity of the PC and PV populations. This analysis was restricted to the lower light intensity (<0.3 mW.mm^-2^), before the spike rate of PV neurons begin to increase. The PCs and PV neurons show proportional decrease in relative spike rate in S1 but not in ALM layer 5 (see above).

b) Further, Figure 7—figure supplement 3 shows that the right parameters yield a great proportional decrease. I believe that this parameter regime is potentially small, but ideally it would be nice to include a figure showing how small (to my knowledge, one like Figure 3—figure supplement 3 doesn't exist for this parameter regime).

Model 1 *can* account for the proportional decrease in S1 provided that the interaction strengths are fine-tuned. As suggested by the reviewers, we now add heatmaps in Figure 7—figure supplement 4 to show how exquisitely the parameters have to be fine-tuned. We modified the text accordingly and write after Figure 7:

“We show in Figure 7—figure supplement 4 that this proportional decrease only happens in a small region of parameter space when the determinant of the interaction matrix, Jαβϵβ, is close to zero.”

The fraction of the different subpopulations of neurons varies across the layers. We do not try to argue that the subpopulations included in Model 1 and Model 2 are the only subpopulations present in ALM and S1. Rather, we propose minimum models that underlie the response properties of these areas assuming that other subpopulations do not qualitatively affect the responses. We make this clearer in the revised manuscript. We write in the Discussion:

“Interneurons are known to be unevenly distributed throughout the cortex. For instance, SOM neurons have been reported to be most prominent in layer 5 whereas VIP neurons are mostly found in layer 2/3 (Tremblay et al., 2016). Instead of giving a complete description of these layers and all neuronal populations they include, we propose here models with the minimal number of inhibitory populations that can account for the data.”

With regard to Model 2, the inhibitory population X we introduced is not necessarily a *specific* subpopulation of inhibitory interneurons. Population X could equally describe the *effective* interaction of several populations with PC and PV. This is now explained in the discussion of Model 2:

“The main difference between Model 1 and Model 2 is that in Model 1, the third inhibitory population (VIP) projects to SOM neurons while in Model 2, the third population (X) does not. […] Alternatively, population X could describe the *effective* interaction of several inhibitory populations with PCs and PV neurons.”

4) Relatedly, the authors propose a series of improved network architectures over the traditional E,I-two population model that incorporate known interneuron subclasses (PV, SOM, and VIP). However, with every improvement the more complex models bring, I find additional questions regarding the data that is not captured.a) For example, Model 1 provides a network that is able to provide a non-paradoxical response such that PV neurons increase in rate at low light stimulations. This is not achievable by the E,I network. However, the heterogeneity of neurons seen as a function of stimulus strength in Model 1 seems different than experimental results (relevant figures for comparison: Figure 1B, Figure 3C, and Figure 5A), and is not discussed in the text. I would've hoped that adding such a large change in the network would've been able to better capture the data.

We agree that the heterogeneity in the model is different from that in the data. We have now discussed this in the manuscript (see our response to 2 above).

b) I would also like to directly compare Figure 2 (left column) with Figure 6 (left or right column). The distribution of variability of firing rates seems different between Model 1 and the experimental results. However, the strength of stimulus is different in Figure 2 than in Figure 6 (0.5 for the experiments than 0.3 and 0.9 for the simulations), so I would request that Figure 2 be remade with one of these stimulation strengths. Having a similar pie chart appear in Figure 2 would also be helpful. Lastly, marginal histograms in both figures would assist with comparison.

We have added pie charts in Figure 2 and now plot in Figure 6, 8 and 9 the responses to 0.5 mW.mm^-2^ (and to 1 mW.mm^-2^).

Also, in regards to Figure 6, the authors comment that "Remarkably, even for 0.9 mW/mm^2^, some of the PCs show an activity increase." However, this was not observed in experiments.

We agree that Model 1 has more heterogeneity than in the experimental data. See, however, our response to comment 2 above.

5) To capture the final missing piece of the data (i.e., the proportional decrease of PC and PV cells), the authors propose an entirely new network architecture with inhibitory interneuron 'X'.a) In addition to this new type of neuron, the authors must also add a connection from PV to SOM cells (otherwise, r_E_ = 0 in the large network limit). While the authors suggest that 'X' may be chandelier cells, they do not discuss why this added connection from PV to SOM would be present in layer 5 but not layer 2/3.

In Model 1 we use the architecture described by (Pfeffer et al., 2013) in which there is no PV to SOM connection. Other studies reported such a connection (Jiang et al., 2015). We did not add this connection to keep the analysis as simple as possible. However, adding PV to SOM connection does not qualitatively affect the responses in Model 1. This is now explained in the Results. We write:

“Following (Pfeffer et al., 2013), the PV population does not project to the SOM population. Other studies have reported such a connection (Jiang et al., 2015). However, adding such a connection to Model 1 does not qualitatively affect the PC and PV responses (see Appendix 1C).”

b) Similar to my above comment, I would expect that such a drastic change in the network would be able to capture additional features of the experimental data, but the heterogeneity present in Figure 10A is drastically different than Figure 1B, right column.

Model 2 is superior to Model 1 since it accounts for the proportionality of the PC and PV response in S1. We agree however, that there is more heterogeneity in our simulations in Model 2 than in the experiments. See our response to comment 2 above. We now mentioned it in the Results section. We write:

“Single neuron responses are more heterogeneous than in the experimental data. It should be noted however that we did not tune parameters to match the experimental heterogeneity.”

We also changed the corresponding section in the Discussion. We now write:

“We observed an even larger diversity in single neuron responses in our simulations of Model 1 and 2.”

Simulations must be made publicly available.

Software will be uploaded to a GitHub repository.